# Robust Data Pruning under Label Noise via Maximizing Re-labeling Accuracy

**Dongmin Park[1], Seola Choi[1], Doyoung Kim[1], Hwanjun Song[2], Jae-Gil Lee[1]***

[1] KAIST, [2] AWS AI Labs

{dongminpark, seola.choi, dodokim, jaegil}@kaist.ac.kr, hwanjun.song@amazon.com

## Abstract

*Data pruning*, which aims to downsize a large training set into a small informative subset, is crucial for reducing the enormous computational costs of modern deep learning. Though large-scale data collections invariably contain annotation noise and numerous robust learning methods have been developed, data pruning for the *noise-robust learning* scenario has received little attention. With state-of-the-art *Re-labeling* methods that self-correct erroneous labels while training, it is challenging to identify which subset induces the most accurate re-labeling of erroneous labels in the entire training set. In this paper, we formalize the problem of *data pruning with re-labeling*. We first show that the likelihood of a training example being correctly re-labeled is proportional to the prediction confidence of its neighborhood in the subset. Therefore, we propose a novel data pruning algorithm, Prune4ReL, that finds a subset maximizing the total neighborhood confidence of all training examples, thereby maximizing the re-labeling accuracy and generalization performance. Extensive experiments on four real and one synthetic noisy datasets show that Prune4ReL outperforms the baselines with Re-labeling models by up to 9.1% as well as those with a standard model by up to 21.6%.

## 1 Introduction

By virtue of ever-growing datasets and the neural scaling law [1, 2], where the model accuracy often increases as a power of the training set size, modern deep learning has achieved unprecedented success in many domains, *e.g.*, GPT [3], CLIP [4], and ViT [5]. With such massive datasets, however, practitioners often suffer from enormous computational costs for training models, tuning their hyper-parameters, and searching for the best architectures, which become the main bottleneck of development cycles. One popular framework to reduce these costs is *data pruning*, which reduces a huge training set into a small subset while preserving model accuracy. Notably, Sorscher et al. [6] have shown that popular data pruning approaches can break down the neural scaling law from power-law to exponential scaling, meaning that one can reach a desired model accuracy with much fewer data. Despite their great success, the impact of *label noise* on data pruning has received little attention, which is unavoidable in real-world data collection [7, 8, 9].

Noisy labels are widely known to severely degrade the generalization capability of deep learning, and thus numerous robust learning strategies have been developed to overcome their negative effect in deep learning [10]. Among them, *Re-labeling* [11], a family of methods that identify wrongly labeled examples and correct their labels during training by a self-correction module such as self-consistency regularization [12], has shown state-of-the-art performance. For example, the performance of DivideMix [13] trained on the CIFAR-10N [7] dataset containing real human annotation noise is nearly identical to that of a standard model trained on the clean CIFAR-10 dataset. Consequently, it is evident that this excellent performance of re-labeling must be carefully considered when designing a framework for data pruning under label noise.

---

*Corresponding author.

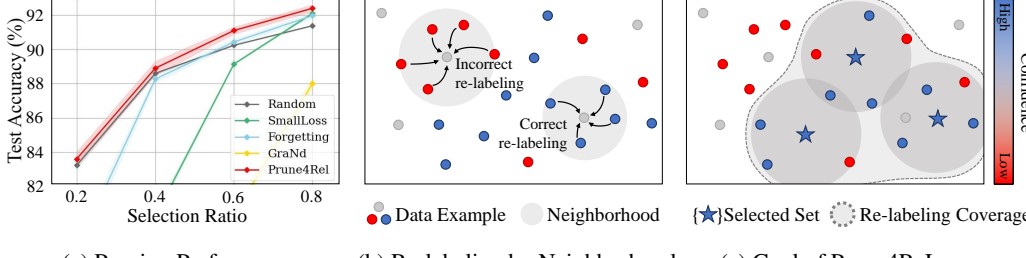

| (a) Pruning Performance. | (b) Re-labeling by Neighborhood. | (c) Goal of Prune4ReL. |

Figure 1: Key idea of Prune4ReL: (a) shows data pruning performance of Prune4ReL and existing sample selection methods on CIFAR-10N with DivideMix; (b) shows how the neighborhood confidence affects the re-labeling correctness; (c) shows the goal of Prune4ReL that maximize the neighbor confidence coverage to the entire training set, thereby maximizing the re-labeling accuracy.

In this paper, we formulate a new problem of *data pruning with re-labeling* for a training set with noisy labels, which aims to maximize the generalization power of the selected subset with expecting that a large proportion of erroneous labels are self-corrected (*i.e.*, re-labeled). Unfortunately, prior data pruning and sample selection algorithms are not suitable for our problem because the re-labeling capability is not taken into account, and have much room for improvement as shown in Figure 1(a). Popular data pruning approaches (denoted as Forgetting [14] and GraNd [15] in blue and yellow, respectively) favor hard (*i.e.*, uncertain) examples because they are considered more beneficial for generalization [16]; however, because it is very difficult to distinguish between hard examples and incorrectly-labeled examples [17], many of the incorrectly-labeled examples can be included in the subset causing unreliable re-labeling. In addition, the small-loss trick [18] (denoted as SmallLoss in green) for sample selection favors easy examples because they are likely to be correctly labeled; however, they are not beneficial for generalization at a later stage of training. Therefore, this new problem necessitates the development of a new data pruning approach.

Accordingly, we suggest a completely novel approach of finding a subset of the training set such that *the re-labeling accuracy of all training examples is preserved as much as possible with the model trained on the subset*. The first challenge in this direction is how to estimate whether each example can be re-labeled correctly even before fully training models on the candidate subset. The second challenge is how to find the subset that maximizes the overall re-labeling accuracy of the entire training set in an efficient manner.

Addressing these two challenges, we develop a novel framework, called **Prune4ReL**. For the first challenge, we define the concept of the *neighborhood confidence* which is the sum of the prediction confidence of each neighbor example in the selected subset. We show that, as in Figure 1(b), an example with high neighborhood confidence is likely to be corrected by Re-labeling methods. We further provide theoretical and empirical evidence of this argument. For the second challenge, we show that the overall re-labeling accuracy is maximized by selecting a subset that maximizes the sum of its reachable neighborhood confidence for all training examples, as shown in Figure 1(c). Furthermore, because enumerating all possible subsets is a combinatorial optimization problem which is NP-hard [19], we provide an efficient greedy selection algorithm that expands the subset one by one by choosing the example that most increases the overall neighborhood confidence.

Extensive experiments on four real noisy datasets, CIFAR-10N, CIFAR-100N, WebVision, and Clothing-1M, and one synthetic noisy dataset on ImageNet-1K show that Prune4ReL consistently outperforms the *eight* data pruning baselines by up to 9.1%. Moreover, Prune4ReL with Re-labeling models significantly outperforms the data pruning baselines with a standard model by up to 21.6%, which reaffirms the necessity of data pruning with re-labeling.

## 2 Preliminary and Related Work

### 2.1 Robust Learning under Noisy Labels

A long line of literature has been proposed to improve the robustness of DNNs against label noise, and refer to [10] for a detailed survey for deep learning with noisy labels. While some studies have

focused on modifying the architectures [20, 21, 22] and the loss functions [23, 24, 25], others have opted for sample selection approaches [26, 18, 27, 28] that select as many clean examples as possible while discarding noisy examples based on a certain cleanness criterion, *e.g.*, small-loss [18]. Note that, these works do not consider the compactness or efficiency of the selected subset. Meanwhile, to further exploit even noisy examples for training, *Re-labeling* [11, 29] approaches try to correct noisy labels and reuse them for training with a re-labeling module, *e.g.*, a heuristic rule [11]. Notably, according to recent benchmark studies on real-world noisy datasets [7], a family of Re-labeling methods with *self-consistency regularization* [12] has shown state-of-the-art performance.

In general, Re-labeling methods with self-consistency regularization are based on

$$\mathcal{L}_{Re\text{-}labeling}(\tilde{\mathcal{D}}; \theta, \mathcal{A}) = \sum_{(x,\tilde{y}) \in \tilde{\mathcal{D}}} \mathbb{1}_{[C_\theta(x) \geq \delta]} \mathcal{L}_{ce}(x, \tilde{y}; \theta) + \lambda \sum_{x \in \tilde{\mathcal{D}}} \mathcal{L}_{reg}(x; \theta, \mathcal{A}), \tag{1}$$

where $\tilde{\mathcal{D}} = \{(x_i, \tilde{y}_i)\}_{i=1}^{m}$ is a given noisy training set obtained from a noisy joint distribution $\mathcal{X} \times \tilde{\mathcal{Y}}$, $\theta$ is a classifier, $\mathcal{A}$ is a strong data augmentation, $C_\theta(\cdot)$ is a prediction confidence score, and $\delta$ is a threshold to identify confident (clean) examples for the supervised loss $\mathcal{L}_{ce}(x, \tilde{y}; \theta)$, *i.e.*, cross-entropy. The noisy labels are implicitly corrected by the self-consistency loss $\mathcal{L}_{reg}(x; \theta)$ leveraging the power of strong augmentations [30]. DivideMix [13], ELR+ [31], CORES [32], and SOP+ [33] are popular approaches belonging to this family. DivideMix uses a co-training framework to further improve the re-labeling accuracy, and SOP+ introduces additional learnable variables combined with a self-consistency loss. For simplicity, we call this Re-labeling family with self-consistency regularization as "Re-labeling" throughout the paper. Despite their effectiveness, Re-labeling models tend to require more computation time due to additional data augmentations, multiple backbones, and longer training epochs, which raises a need to enhance its efficiency.

## 2.2 Data Pruning

In order to achieve high generalization performance with a selected subset, most data pruning approaches often prioritize the selection of hard or uncertain examples. Specifically, *uncertainty-based* methods [34, 35, 36] favor selecting less confident examples over more confident ones, as the former is assumed to be more informative than the latter. Similarly, *geometry-based* methods [37, 38] focus on removing redundant examples that are close to each other in the feature space, and *loss-based* methods [14, 15, 39, 40] favor selecting the examples with a high loss or gradient measured during training. However, these existing methods may not be effective in realistic scenarios under label noise because noisy examples also exhibit high uncertainty and could be wrongly considered informative for training [14]. Meanwhile, some recent works reported that existing data pruning methods do not work well at high pruning ratios [41, 42]. To alleviate this drawback, AL4DP [41] shows that mixing various levels of uncertain examples is better for data scarcity, Moderate [43] aims to select examples with the distances close to the median, and CCS [42] proposes a coverage-based method that jointly considers data coverage with sample importance. While a few works attempted to improve the robustness of sample selection against label noise by filtering the noise [19], no work yet considers the effect of data pruning on noise-robust learners such as Re-labeling.

## 3 Methodology

We formalize a problem of data pruning with re-labeling such that it finds the most informative subset $\mathcal{S}$, where a model $\theta_\mathcal{S}$ trained on $\mathcal{S}$ maximizes the re-labeling accuracy of the entire noisy training set $\tilde{\mathcal{D}} = \{(x_i, \tilde{y}_i)\}_{i=1}^{m}$ [^2]. Formally, we aim to find an optimal subset $\mathcal{S}^*$ such that

$$\mathcal{S}^* = \underset{\mathcal{S}: \, |\mathcal{S}| \leq s}{\operatorname{argmax}} \sum_{(x,\tilde{y}) \in \tilde{\mathcal{D}}} \mathbb{1}_{[f(x; \theta_\mathcal{S}) \, = \, y^*]} \quad : \quad \theta_\mathcal{S} = \underset{\theta}{\operatorname{argmin}} \; \mathcal{L}_{Re\text{-}labeling}(\mathcal{S}; \theta, \mathcal{A}), \tag{2}$$

where $y^*$ is the ground-truth label of a noisy example $x$, $f(x; \theta_\mathcal{S}) \in \mathbb{R}^c$ is a $c$-way class prediction of the example $x$ from the Re-labeling model $\theta_\mathcal{S}$, and $s$ is the target subset size.

Finding the optimal subset $\mathcal{S}^*$ through direct optimization of Eq. (2) is infeasible because the ground-truth label $y^*$ is unknown in practice. In addition, the subset should be found at the early stage of

---

[^2]: Maximizing the re-labeling accuracy is equivalent to correctly re-labeling all training examples. This formulation enables the model to exploit all clean labels for training, leading to a satisfactory generalization.

training, *i.e.*, in a warm-up period, to reduce the computational cost [16]. To achieve these goals in an accurate and efficient way, in Section 3.1, we first introduce a new metric, *the reduced neighborhood confidence*, that enables estimating the re-labeling capacity of a subset even in the warm-up period. Then, in Section 3.2, we propose a new data pruning algorithm *Prune4ReL* using this reduced neighborhood confidence to find a subset that maximizes the re-labeling accuracy.

## 3.1 Reduced Neighborhood Confidence

As a measurement of estimating the re-labeling accuracy, we use the confidence of neighbor examples for each target noisy example $x$, because the noisy examples are known to be corrected by their *clean neighbor* examples with self-consistency regularization [44]. Specifically, once an augmentation of a noisy example has a similar embedding to those of other clean neighbors in the representation space, the self-consistency loss can force the prediction of the noisy example to be similar to those of other clean neighbors as a way of re-labeling. This property is also evidenced by a theory of re-labeling with a generalization bound [45]. Thus, the neighboring relationship among examples can be a clear clue to estimate the re-labeling accuracy even in the early stage of training.

We define a *neighborhood* and its *reduced neighborhood confidence* to utilize the relationship of neighboring examples in Definitions 3.1 and 3.2.

**Definition 3.1.** (NEIGHBORHOOD). Let $\mathcal{B}(x_i) = \{x : ||\mathcal{A}(x_i) - x|| \leq \epsilon\}$ be a set of all possible augmentations from the original example $x_i$ using an augmentation function $\mathcal{A}$. Then, given a noisy training set $\tilde{\mathcal{D}}$, a *neighborhood* of $x_i$ is defined as $\mathcal{N}(x_i) = \{x \in \tilde{\mathcal{D}} : \mathcal{B}(x_i) \cap \mathcal{B}(x) \neq \emptyset\}$, which is the set of examples that are reachable by the augmentation $\mathcal{A}$. □

**Definition 3.2.** (REDUCED NEIGHBORHOOD CONFIDENCE). The *reduced neighborhood confidence* $C_{\mathcal{N}}(x_i; \mathcal{S})$ of an example $x_i$ is the sum of the prediction confidence $C(\cdot)$ of its neighbors $x_j \in \mathcal{N}(x_i)$ in a given reduced (*i.e.*, selected) subset $\mathcal{S}$, which is formalized as

$$C_{\mathcal{N}}(x_i; \mathcal{S}) = \sum_{x_j \in \mathcal{S}} \mathbb{1}_{[x_j \in \mathcal{N}(x_i)]} \cdot C(x_j), \tag{3}$$

and its *empirical reduced neighborhood confidence* is computed by using the cosine similarity among the augmentations of all possible pairs of example embeddings,

$$\hat{C}_{\mathcal{N}}(x_i; \mathcal{S}) = \sum_{x_j \in \mathcal{S}} \mathbb{1}_{[sim(\mathcal{A}(x_i), \mathcal{A}(x_j)) \geq \tau]} \cdot sim\big(\mathcal{A}(x_i), \mathcal{A}(x_j)\big) \cdot C(x_j), \tag{4}$$

where $sim(\cdot)$ is the cosine similarity between the augmentations $\mathcal{A}(x)$ of two different examples in the embedding space, and $\tau$ a threshold to determine whether the two examples belong to the same neighborhood. Unlike Eq. (3), Eq. (4) is calculated as a weighted sum of prediction confidences with cosine similarity to approximate the likelihood of belonging to the neighborhood. □

Based on these definitions, we investigate the theoretical evidence of employing the reduced neighborhood confidence as a means to estimate the re-labeling capacity of a subset.

**Theoretical Evidence.** A subset $\mathcal{S}$ with a *high* value of the *total* reduced neighborhood confidence, the sum of the reduced neighborhood confidence of each example in $\mathcal{S}$, allows a Re-labeling model to maximize its re-labeling accuracy in the entire training set. We formally support this optimization by providing a theoretical analysis that extends the generalization bound in the prior re-labeling theory [45] to data pruning.

**Assumption 3.3.** (EXPANSION AND SEPARATION). Following the assumption in [45], the $\alpha$-expansion and $\beta$-separation assumptions hold for the training set $\tilde{\mathcal{D}}$. The $\alpha$-expansion means that an example is reachable to the $\alpha$ number of augmentation neighbors on average, *i.e.*, $\mathbb{E}_{x \in \tilde{\mathcal{D}}}[|\mathcal{N}(x)|] = \alpha$. The $\beta$-separation means that data distributions with different ground-truth classes are highly separable, such that the average proportion of the neighbors from different classes is as small as $\beta$.

Under these assumptions, we can obtain a training accuracy (error) bound of a Re-labeling model trained on a subset $S$ as in Theorem 3.4.

**Theorem 3.4.** *Assume that a subset $\mathcal{S} \in \tilde{\mathcal{D}}$ follows $\alpha_{\mathcal{S}}$-expansion and $\beta_{\mathcal{S}}$-separation, where $\alpha_{\mathcal{S}} \leq \alpha$. Then, the training error of a Re-labeling model $\theta_{\mathcal{S}}$ trained on $\mathcal{S}$ is bounded by the inverse of the total reduced neighborhood confidence $\sum_{x \in \tilde{\mathcal{D}}} C_{\mathcal{N}}(x; \mathcal{S})$ such that,*

$$Err(\theta_{\mathcal{S}}) \leq \frac{2 \cdot |\mathcal{S}| \cdot Err(\theta_{\mathcal{M}})}{\sum_{x \in \tilde{\mathcal{D}}} C_{\mathcal{N}}(x; \mathcal{S})} + \frac{2 \cdot \alpha_{\mathcal{S}}}{\alpha_{\mathcal{S}} - 1} \cdot \beta_{\mathcal{S}}, \tag{5}$$

*where $\theta_{\mathcal{M}}$ is a model trained with the supervised loss in Eq. (1) on a given clean set $\mathcal{M} \subset \mathcal{S}$.*

*Proof.* We extend the label denoising theorem in [45] by incorporating the influence of the subset to the expansion factor $\alpha_{\mathcal{S}}$. The complete proof is available in Appendix A. □

Since $\beta_{\mathcal{S}}$ is usually very small, its effect on the error bound is negligible. Then, the bound highly depends on the total reduced neighborhood confidence. That is, as the total reduced neighborhood confidence increases, the error bound becomes tighter. This theorem supports that we can utilize the reduced neighborhood confidence for the purpose of maximizing the re-labeling accuracy.

**Empirical Evidence.** To empirically support Theorem 3.4, we validate the correlation between the empirical reduced neighborhood confidence[3] and the re-labeling accuracy using CIFAR-10N, which is a real-world noisy benchmark dataset.

Specifically, we train DivideMix [13] on the 20% randomly selected subset $\mathcal{S}$ for a warm-up training epoch of 10 and calculate the empirical reduced neighborhood confidence $\hat{C}_{\mathcal{N}}(x; \mathcal{S})$ for the entire training set. Next, we fully train DivideMix [13] on the random subset $\mathcal{S}$. Last, we divide the entire training set into 15 bins according to the obtained $\hat{C}_{\mathcal{N}}(x; \mathcal{S})$ and verify the average re-labeling accuracy for each bin.

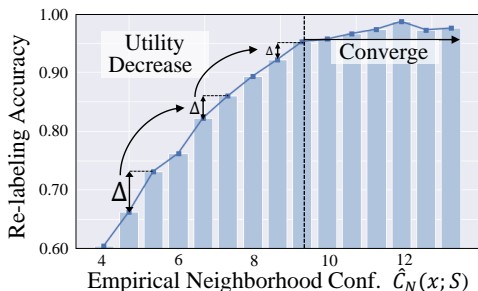

Figure 2: Correlation between neighborhood confidence and re-labeling accuracy on a 20% randomly selected subset of CIFAR-10N.

Figure 2 shows how the re-labeling accuracy changes according to the empirical reduced neighborhood confidence in Eq. (4). (The term "empirical" is simply omitted hereafter.) The re-labeling accuracy shows a strong positive correlation with the reduced neighborhood confidence. Interestingly, as the neighborhood confidence increases, its *utility* in improving the re-labeling accuracy decreases, eventually reaching a convergence point after surpassing a certain threshold.

### 3.2 Data Pruning by Maximizing Neighborhood Confidence Coverage

We present a new data pruning algorithm called *Prune4ReL* which optimizes the total reduced neighborhood confidence defined in Eq. (4). This objective is equivalent to identifying a subset that maximizes the re-labeling accuracy on the entire training set, as justified in Theorem 3.4. Therefore, the objective of Prune4ReL is to find the subset $\mathcal{S}^*$, which is formulated as

$$\mathcal{S}^* = \underset{\mathcal{S}: |\mathcal{S}| \leq s}{\arg\max} \sum_{x_i \in \tilde{\mathcal{D}}} \boldsymbol{\sigma}\big(\hat{C}_{\mathcal{N}}(x_i; \mathcal{S})\big), \tag{6}$$

where $\boldsymbol{\sigma}(z)$ is a *utility* function of the reduced neighborhood confidence $\hat{C}_{\mathcal{N}}(x_i; \mathcal{S})$ in improving the re-labeling accuracy. By the observation in Figure 2, we define $\boldsymbol{\sigma}(z)$ as a *non-decreasing* and *concave* function where $\boldsymbol{\sigma}(0) = 0$. In our implementation, we use the positive of the $tanh$ function as the utility function, *i.e.*, $\boldsymbol{\sigma}(z) = tanh(z)$. However, directly solving Eq. (6) is computationally expensive and impractical due to its NP-hard nature as a Set-Cover problem [19]. Accordingly, we employ an approximation solution to efficiently address this combinatorial optimization.

**Optimization with Greedy Approximation.** We present a practical solution for solving the optimization problem stated in Eq. (6) using a greedy approximation. The objective function satisfies both the *monotonicity* and *submodularity* conditions, indicating that the return of the objective function monotonically increases and the marginal benefit of adding an example decreases as the subset grows. Thus, a greedy sample selection can be employed as in Algorithm 1.

---

[3]RandAug [30] is used as the augmentation function for the reduced neighborhood confidence.

In detail, we begin with an empty set $\mathcal{S}$ and initialize the reduced neighborhood confidence $\hat{C}_{\mathcal{N}}$ to 0 (lowest confidence) for all training examples (Line 1). Next, at every step, we select an example $x$ that maximizes the marginal benefit $\boldsymbol{\sigma}(\hat{C}_{\mathcal{N}}(x) + C(x)) - \boldsymbol{\sigma}(\hat{C}_{\mathcal{N}}(x))$ of Eq. (6), and update the reduced neighborhood confidence $\hat{C}_{\mathcal{N}}$ based on the similarity scores (Lines 3–7). To further improve robustness and efficiency, we introduce a class-balanced version, Prune4ReL$_B$, of which the detailed process is elaborated in Appendix B.

---

**Algorithm 1** Greedy Neighborhood Confidence

INPUT: $\tilde{\mathcal{D}}$: training set, $s$: target subset size, and $C(x)$: confidence from warm-up classifier
1: Initialize $\mathcal{S} \leftarrow \emptyset; \forall x \in \tilde{\mathcal{D}}, \hat{C}_{\mathcal{N}}(x) = 0$
2: **repeat**
3:    $x = \mathrm{argmax}_{x \in \tilde{\mathcal{D}} \setminus \mathcal{S}} \boldsymbol{\sigma}(\hat{C}_{\mathcal{N}}(x) + C(x)) - \boldsymbol{\sigma}(\hat{C}_{\mathcal{N}}(x))$
4:    $\mathcal{S} = \mathcal{S} \cup \{x\}$
5:    **for all** $v \in \tilde{\mathcal{D}}$ **do**
6:      $\hat{C}_{\mathcal{N}}(v) \mathrel{+}= \mathbb{1}_{[sim(x,v) \geq \tau]} \cdot sim(x,v) \cdot C(x)$
7: **until** $|\mathcal{S}| = s$
OUTPUT: Final selected subset $\mathcal{S}$

---

In Theorem 3.5, we guarantee the selected subset $\mathcal{S}$ obtained by our greedy solution achieves a $(1 - 1/e)$-approximation of the optimum.

**Theorem 3.5.** *Since Eq.* (6)*, denoted as $OBJ$, is a monotone, submodular, and non-negative function on $x$, the greedy solution provides a set with a $(1 - 1/e)$-approximation of the optimum. Formally,*

$$OBJ(\mathcal{S}) \geq (1 - 1/e) \cdot OBJ(\mathcal{S}^*). \tag{7}$$

*Proof.* We prove the monotonicity and submodularity of Eq. (6). If the two conditions are satisfied, Eq. (7) naturally holds. See Appendix C for the complete proof. □

**Time Complexity Analysis.** We analyze the time complexity of our greedy approximation in Algorithm 1. At each step, Prune4ReL takes the time complexity of $O(m \log m) + O(md)$, where $m$ is the training set size and $d$ is the embedding dimension size of the warm-up classifier. Specifically, in Line 3, sampling an example with the largest marginal benefit of confidence takes $O(m \log m)$, and in Lines 5–6, updating the reduced neighborhood confidence of all training examples takes $O(md)$. In addition, with Prune4ReL$_B$, the time complexity is reduced to $O(m \log(m/c)) + O(md)$ because it iteratively selects the example with the largest marginal benefit within each class subset, which is lower than the time complexity of a similar distance-based data pruning work, kCenterGreedy [38] aiming to maximize the *distance* coverage of a selected subset to the entire training set by a greedy approximation. At each iteration, kCenterGreedy's runtime is $O(mk_t)$, where $k_t$ is the size of the selected set at iteration $t$ [46]. Note that, its time complexity increases as the subset size grows, which hinders its usability on a large-scale dataset. In Section 4.2, we empirically show that Prune4ReL is scalable to prune Clothing-1M, a large dataset with 1M examples, whereas kCenterGreedy is not.

## 4 Experiments

### 4.1 Experiment Setting

**Datasets.** We first perform the data pruning task on four *real* noisy datasets, CIFAR-10N, CIFAR-100N, Webvision, and Clothing-1M. CIFAR-10N and CIFAR-100N [7] contain human re-annotations of 50K training images in the original CIFAR-10 and CIFAR-100 [47]. Specifically, each training image in CIFAR-10N contains three noisy labels, called Random 1,2,3, which are further transformed into the Worst-case label. Each image in CIFAR-100N contains one noisy label. WebVision [8] and Clothing-1M [9] are two large-scale noisy datasets. WebVision contains 2.4M images crawled from the Web using the 1,000 concepts in ImageNet-1K [48]. Following prior work [49], we use mini-WebVision consisting of the first 50 classes of the Google image subset with approximately 66K training images. Clothing-1M consists of 1M training images with noisy labels and 10K clean test images collected from online shopping websites. Additionally, a large-scale *synthetic* noisy dataset, which we call ImageNet-N, is included in our experiments. It consists of 1.2M training images, which are the training images of ImageNet-1K [48] with asymmetric label noise. See Appendix D for details of the noise injection.

**Algorithms.** We compare Prune4ReL with a random selection from a uniform distribution, Uniform, a clean sample selection algorithm, SmallLoss [26], and six data pruning algorithms including Margin [50], $k$-CenterGreedy [38], Forgetting [14], GraNd [15], SSP [6], and Moderate [43]. SmallLoss favors examples with a small loss. For data pruning algorithms, (1) Margin selects examples in the

Table 1: Performance comparison of sample selection baselines and Prune4ReL on CIFAR-10N and CIFAR-100N. The best results are in bold.

| Re-label Models | Selection Methods | CIFAR-10N | | | | | | | | CIFAR-100N | | | |
| --- | --- | --- | --- | --- | --- | --- | --- | --- | --- | --- | --- | --- | --- |
| | | Random | | | | Worst | | | | Noisy | | | |
| | | 0.2 | 0.4 | 0.6 | 0.8 | 0.2 | 0.4 | 0.6 | 0.8 | 0.2 | 0.4 | 0.6 | 0.8 |
| DivMix | Uniform | $87.5_{\pm0.2}$ | $91.9_{\pm0.2}$ | $93.7_{\pm0.1}$ | $94.8_{\pm0.1}$ | $83.2_{\pm0.2}$ | $88.5_{\pm0.1}$ | $90.2_{\pm0.0}$ | $91.4_{\pm0.0}$ | $30.5_{\pm1.0}$ | $55.3_{\pm0.5}$ | $57.5_{\pm1.9}$ | $58.6_{\pm0.9}$ |
| | SmallL | $68.1_{\pm4.0}$ | $82.4_{\pm0.8}$ | $89.0_{\pm0.3}$ | $93.1_{\pm0.1}$ | $70.3_{\pm0.6}$ | $80.3_{\pm0.2}$ | $89.1_{\pm0.0}$ | $92.1_{\pm0.1}$ | $33.3_{\pm3.2}$ | $47.4_{\pm1.1}$ | $59.4_{\pm0.7}$ | $62.0_{\pm1.2}$ |
| | Margin | $68.5_{\pm2.9}$ | $88.5_{\pm0.3}$ | $93.2_{\pm0.2}$ | $94.7_{\pm0.1}$ | $61.3_{\pm0.8}$ | $75.1_{\pm0.7}$ | $85.3_{\pm0.2}$ | $90.2_{\pm0.1}$ | $17.1_{\pm0.8}$ | $30.8_{\pm1.0}$ | $46.3_{\pm2.4}$ | $61.2_{\pm1.3}$ |
| | kCenter | $87.4_{\pm0.5}$ | $\mathbf{93.0}_{\pm0.1}$ | $94.4_{\pm0.1}$ | $95.0_{\pm0.0}$ | $82.7_{\pm0.8}$ | $88.4_{\pm0.1}$ | $90.6_{\pm0.1}$ | $92.2_{\pm0.1}$ | $38.0_{\pm1.0}$ | $50.0_{\pm1.5}$ | $59.7_{\pm1.3}$ | $63.0_{\pm0.9}$ |
| | Forget | $85.2_{\pm0.5}$ | $\mathbf{93.0}_{\pm0.1}$ | $\mathbf{94.5}_{\pm0.1}$ | $\mathbf{95.1}_{\pm0.1}$ | $78.3_{\pm0.6}$ | $88.3_{\pm0.2}$ | $90.4_{\pm0.1}$ | $92.0_{\pm0.2}$ | $26.4_{\pm1.3}$ | $54.3_{\pm0.8}$ | $63.1_{\pm1.3}$ | $66.6_{\pm1.1}$ |
| | GraNd | $21.8_{\pm0.6}$ | $60.9_{\pm4.5}$ | $92.5_{\pm1.8}$ | $94.8_{\pm0.1}$ | $18.5_{\pm1.7}$ | $25.5_{\pm0.9}$ | $49.3_{\pm0.9}$ | $88.0_{\pm0.5}$ | $15.5_{\pm1.2}$ | $26.0_{\pm1.9}$ | $44.7_{\pm1.5}$ | $60.4_{\pm1.6}$ |
| | SSP | $85.8_{\pm1.8}$ | $92.2_{\pm1.5}$ | $93.0_{\pm1.1}$ | $94.5_{\pm0.2}$ | $81.4_{\pm2.5}$ | $86.5_{\pm1.9}$ | $89.6_{\pm1.2}$ | $91.9_{\pm0.4}$ | $30.1_{\pm2.2}$ | $52.4_{\pm1.3}$ | $58.7_{\pm0.5}$ | $63.4_{\pm0.5}$ |
| | Moderate | $86.4_{\pm0.8}$ | $91.4_{\pm0.3}$ | $93.8_{\pm0.5}$ | $94.8_{\pm0.2}$ | $81.4_{\pm1.2}$ | $86.5_{\pm0.6}$ | $90.0_{\pm0.6}$ | $91.6_{\pm0.2}$ | $34.2_{\pm1.4}$ | $54.5_{\pm1.3}$ | $56.1_{\pm0.5}$ | $59.9_{\pm0.6}$ |
| | **Pr4ReL** | $\mathbf{87.6}_{\pm0.3}$ | $92.4_{\pm0.4}$ | $\mathbf{94.5}_{\pm0.2}$ | $\mathbf{95.1}_{\pm0.1}$ | $\mathbf{83.7}_{\pm0.5}$ | $\mathbf{88.8}_{\pm0.3}$ | $90.2_{\pm0.3}$ | $92.0_{\pm0.3}$ | $37.2_{\pm1.0}$ | $55.3_{\pm0.7}$ | $61.2_{\pm0.5}$ | $65.5_{\pm0.8}$ |
| | **Pr4ReL**$_B$ | $\mathbf{88.1}_{\pm0.3}$ | $\mathbf{93.0}_{\pm0.2}$ | $\mathbf{94.5}_{\pm0.2}$ | $\mathbf{95.1}_{\pm0.1}$ | $\mathbf{83.7}_{\pm0.4}$ | $88.6_{\pm0.4}$ | $\mathbf{90.8}_{\pm0.2}$ | $\mathbf{92.4}_{\pm0.2}$ | $\mathbf{39.4}_{\pm0.8}$ | $\mathbf{56.3}_{\pm0.5}$ | $\mathbf{63.5}_{\pm0.3}$ | $\mathbf{67.4}_{\pm0.7}$ |
| SOP+ | Uniform | $87.5_{\pm0.3}$ | $91.5_{\pm0.1}$ | $93.4_{\pm0.0}$ | $94.8_{\pm0.2}$ | $81.9_{\pm0.1}$ | $87.5_{\pm0.1}$ | $90.8_{\pm0.1}$ | $91.8_{\pm0.1}$ | $46.5_{\pm0.0}$ | $55.7_{\pm0.2}$ | $60.8_{\pm0.3}$ | $64.4_{\pm0.2}$ |
| | SmallL | $77.6_{\pm2.5}$ | $86.2_{\pm0.1}$ | $90.7_{\pm0.6}$ | $94.3_{\pm0.2}$ | $78.8_{\pm0.2}$ | $84.1_{\pm0.1}$ | $89.3_{\pm0.1}$ | $92.3_{\pm0.2}$ | $48.5_{\pm0.8}$ | $59.8_{\pm0.4}$ | $63.9_{\pm0.2}$ | $66.1_{\pm0.6}$ |
| | Margin | $52.1_{\pm5.0}$ | $79.6_{\pm8.6}$ | $92.6_{\pm3.9}$ | $95.1_{\pm1.3}$ | $45.7_{\pm1.1}$ | $61.8_{\pm0.7}$ | $84.6_{\pm0.3}$ | $92.5_{\pm0.0}$ | $20.0_{\pm1.2}$ | $34.4_{\pm0.3}$ | $50.4_{\pm0.6}$ | $63.3_{\pm0.1}$ |
| | kCenter | $86.3_{\pm0.4}$ | $92.2_{\pm0.3}$ | $94.1_{\pm0.2}$ | $\mathbf{95.3}_{\pm0.1}$ | $81.9_{\pm0.0}$ | $88.0_{\pm0.0}$ | $\mathbf{91.3}_{\pm0.1}$ | $92.3_{\pm0.0}$ | $44.8_{\pm0.6}$ | $55.9_{\pm0.4}$ | $61.6_{\pm0.3}$ | $65.2_{\pm0.6}$ |
| | Forget | $82.4_{\pm1.0}$ | $93.0_{\pm0.2}$ | $94.2_{\pm0.3}$ | $95.0_{\pm0.1}$ | $71.1_{\pm0.4}$ | $87.7_{\pm0.1}$ | $90.6_{\pm0.3}$ | $92.2_{\pm0.0}$ | $38.0_{\pm0.5}$ | $55.3_{\pm0.2}$ | $63.2_{\pm0.1}$ | $65.8_{\pm0.4}$ |
| | GraNd | $24.2_{\pm5.5}$ | $51.6_{\pm3.2}$ | $85.9_{\pm1.2}$ | $94.9_{\pm0.2}$ | $15.4_{\pm1.6}$ | $25.7_{\pm0.6}$ | $51.0_{\pm0.5}$ | $86.8_{\pm0.5}$ | $11.0_{\pm0.1}$ | $19.0_{\pm0.6}$ | $38.7_{\pm1.5}$ | $62.1_{\pm0.5}$ |
| | SSP | $80.5_{\pm2.6}$ | $91.7_{\pm1.5}$ | $93.8_{\pm1.0}$ | $95.0_{\pm0.2}$ | $70.8_{\pm2.7}$ | $86.6_{\pm1.9}$ | $89.2_{\pm0.9}$ | $92.3_{\pm0.4}$ | $39.2_{\pm2.2}$ | $54.9_{\pm1.5}$ | $62.7_{\pm0.7}$ | $65.0_{\pm0.3}$ |
| | Moderate | $87.8_{\pm1.0}$ | $92.8_{\pm0.5}$ | $94.0_{\pm0.3}$ | $94.9_{\pm0.2}$ | $75.2_{\pm1.5}$ | $81.9_{\pm1.2}$ | $87.7_{\pm0.7}$ | $91.8_{\pm0.3}$ | $46.4_{\pm1.8}$ | $54.6_{\pm1.7}$ | $60.2_{\pm0.4}$ | $64.6_{\pm0.4}$ |
| | **Pr4ReL** | $87.8_{\pm1.2}$ | $92.7_{\pm0.4}$ | $\mathbf{94.4}_{\pm0.2}$ | $95.1_{\pm0.1}$ | $82.7_{\pm0.5}$ | $88.1_{\pm0.4}$ | $\mathbf{91.3}_{\pm0.3}$ | $92.5_{\pm0.2}$ | $50.2_{\pm0.2}$ | $59.1_{\pm0.5}$ | $63.9_{\pm0.3}$ | $65.7_{\pm0.5}$ |
| | **Pr4ReL**$_B$ | $\mathbf{88.5}_{\pm0.3}$ | $\mathbf{93.1}_{\pm0.2}$ | $\mathbf{94.4}_{\pm0.1}$ | $\mathbf{95.3}_{\pm0.1}$ | $\mathbf{84.9}_{\pm0.6}$ | $\mathbf{89.2}_{\pm0.6}$ | $\mathbf{91.3}_{\pm0.3}$ | $\mathbf{92.9}_{\pm0.1}$ | $\mathbf{52.9}_{\pm0.8}$ | $\mathbf{60.1}_{\pm0.6}$ | $\mathbf{64.1}_{\pm0.4}$ | $\mathbf{66.2}_{\pm0.3}$ |

increasing order of the difference between the highest and the second highest softmax probability; (2) $k$-CenterGreedy selects $k$ examples that maximize the distance coverage to the entire training set; (3) Forgetting selects examples that are easy to be forgotten by the classifier throughout the warm-up training epochs; (4) GraNd uses the average norm of the gradient vectors to measure the contribution of each example to minimizing the training loss; (5) SSP leverages a self-supervised pre-trained model to select the most prototypical examples; and (6) Moderate aims to select moderately hard examples using the distances to the median.

**Implementation Details.** We train two representative Re-labeling models, DivideMix [13] and SOP+ [33] for our experiments. The hyperparameters for DivideMix and SOP+ are favorably configured following the original papers. Following the prior Re-labeling work [13, 33], for CIFAR-10N and CIFAR-100N, PreAct Resnet-18 [51] is trained for 300 epochs using SGD with a momentum of 0.9, a weight decay of 0.0005, and a batch size of 128. The initial learning rate is 0.02, and it is decayed with a cosine annealing scheduler. For WebVision, InceptionResNetV2 [52] is trained for 100 epochs with a batch size of 32. For Clothing-1M, we use ResNet-50 [53] pre-trained on ImageNet and fine-tune it for 10 epochs with a batch size of 32. The initial learning rates of WebVision and Clothing-1M are 0.02 and 0.002, which are dropped by a factor of 10 at the halfway point of the training epochs. For ImageNet-N, ResNet-50 [53] is trained for 50 epochs with a batch size of 64 and an initial learning rate of 0.02 decayed with a cosine annealing scheduler.

For data pruning algorithms, following prior work [16], we perform sample selection after 10 warm-up training epochs for CIFAR-10N, WebVision, and ImageNet-N, and 30 warm-up epochs for CIFAR-100N. For Clothing-1M, we perform the sample selection after 1 warm-up training epoch from the ImageNet pre-trained ResNet-50. The hyperparameters for all data pruning methods are favorably configured following the original papers. For Prune4ReL, we set its hyperparameter $\tau$ to 0.975 for CIFAR-10N, to 0.95 for CIFAR-100N, WebVision, and ImageNet-N, and to 0.8 for Clothing-1M. More implementation details can be found in Appendix E. All methods are implemented with PyTorch 1.8.0 and executed on NVIDIA RTX 3080 GPUs. The code is available at `https://github.com/kaist-dmlab/Prune4Rel`.

**Evaluation.** For CIFAR-10N, CIFAR-100N, and WebVision, we select the subset with the selection ratios {0.2, 0.4, 0.6, 0.8}. For Clothing-1M and ImageNet-N, we construct the subset with {0.01, 0.05, 0.1, 0.2} and {0.05, 0.1, 0.2, 0.4} selection ratios, respectively. We measure the test accuracy of the Re-labeling models trained from scratch on the selected subset. Every experiment is run three times, and the average of the last accuracy is reported. For CIFAR-10N with the Random noise, we average the test accuracy of the models trained using the three noisy labels.

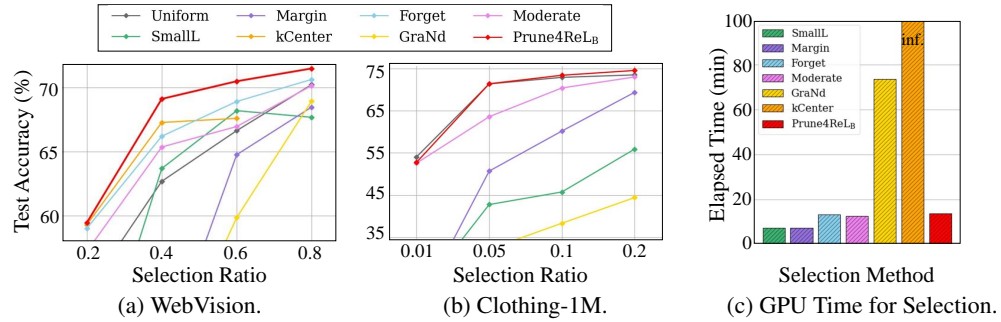

Figure 3: Data pruning performance comparison: (a) test accuracy of SOP+ trained on each selected subset of WebVision; (b) test accuracy of DivideMix trained on each selected subset of Clothing-1M; (c) elapsed GPU time for selecting a subset on WebVision with a selection ratio of 0.8.

Table 2: Performance comparison of the standard cross-entropy model and Re-labeling models when combined with data pruning methods on CIFAR-10N and CIFAR-100N.

| Learning Models | Selection Methods | CIFAR-10N | | | | | | | | CIFAR-100N | | | |
| | | Random | | | | Worst | | | | Noisy | | | |
| | | 0.2 | 0.4 | 0.6 | 0.8 | 0.2 | 0.4 | 0.6 | 0.8 | 0.2 | 0.4 | 0.6 | 0.8 |
| --- | --- | --- | --- | --- | --- | --- | --- | --- | --- | --- | --- | --- | --- |
| Standard | Uniform | 75.6±0.1 | 81.0±0.1 | 83.0±0.1 | 84.3±0.5 | 58.6±0.3 | 63.9±0.2 | 66.4±0.1 | 67.5±0.5 | 37.4±0.2 | 46.1±0.2 | 50.1±0.0 | 52.3±0.1 |
| | SmallL | 75.0±1.5 | 83.4±0.6 | 87.5±0.3 | 90.1±0.2 | 70.1±0.3 | 77.5±1.6 | 80.6±0.5 | 76.4±0.1 | 42.2±0.4 | 54.7±0.5 | 57.7±0.1 | 57.4±0.2 |
| | Forget | 82.2±0.5 | 86.2±0.1 | 86.1±0.0 | 85.4±0.2 | 71.2±0.3 | 73.3±0.2 | 71.4±0.2 | 69.6±0.1 | 43.5±0.4 | 54.5±0.1 | 57.5±0.4 | 56.6±0.3 |
| DivMix | **Pr4ReL**$_B$ | 88.1±0.3 | 93.0±0.2 | 94.5±0.2 | 95.1±0.1 | 83.7±0.4 | 88.6±0.4 | 90.8±0.2 | 92.4±0.2 | 39.4±0.8 | **56.3**±0.5 | **63.5**±0.3 | **67.4**±0.7 |
| SOP+ | **Pr4ReL**$_B$ | 88.5±0.3 | 93.1±0.2 | 94.4±0.1 | 95.3±0.1 | 84.9±0.6 | 89.2±0.6 | 91.3±0.3 | 92.9±0.1 | 52.9±0.8 | 60.1±0.6 | 64.1±0.4 | 66.2±0.3 |

## 4.2 Main Results on Real Noisy Datasets

**Test Accuracy.** Table 1 summarizes the test accuracy of *eight* baselines and Prune4ReL on CIFAR-10N and CIFAR-100N trained with two Re-labeling models. Overall, Prune4ReL consistently achieves the best performance for all datasets across varying selection ratios. Numerically, Prune4ReL improves DivideMix and SOP+ by up to 3.7% and 9.1%, respectively. Compared with six data pruning baselines, they show rapid performance degradation as the size of the subset decreases; most of them, which are designed to favor hard examples, tend to select a large number of noisy examples, resulting in unreliable re-labeling. While Moderate selects moderately hard examples, it is still worse than Prune4ReL since it is not designed for noise-robust learning scenarios with Re-labeling models. On the other hand, SmallLoss, a clean sample selection baseline, shows poor performance in CIFAR-10N (Random), because this dataset contains a relatively low noise ratio, and selecting clean examples is less critical. Although SmallLoss shows robust performance in CIFAR-100N, it is worse than Prune4ReL because it loses many informative noisy examples that help generalization if re-labeled correctly. Meanwhile, Uniform is a fairly robust baseline as it selects easy (clean) and hard (noisy) examples in a balanced way; many selected noisy examples may be relabeled correctly by other selected clean neighbors, resulting in satisfactory test accuracy.

Similarly, Figures 3(a) and 3(b) visualize the efficacy of the baselines and Prune4ReL on the WebVision and Clothing-1M datasets. We train SOP+ on WebVision and DivideMix on Clothing-1M. Similar to CIFAR-N datasets, Prune4ReL achieves better performance than existing baselines on two datasets. Quantitatively, Prune4ReL outperforms the existing sample selection methods by up to 2.7% on WebVision with a selection ratio of 0.4. This result confirms that the subset selected by Prune4ReL, which maximizes the total neighborhood confidence of the training set, successfully maintains the performance of Re-labeling models and is effective for model generalization.

**Efficiency.** In Figure 3(c), we further show the GPU time taken for selecting subsets within the warm-up training. We train SOP+ on WebVision with a selection ratio of 0.8. Powered by our efficient greedy approximation, Prune4ReL fastly prunes the dataset in a reasonable time. GraNd takes almost 10 times longer than SmallLoss or Margin, since it trains the warm-up classifier multiple times for the ensemble. kCenterGreedy is infeasible to run in our GPU configuration due to its huge computation and memory costs.

Table 3: Effect of the confidence metrics on Prune4ReL.

| Re-label Model | Dataset | Conf. Metric | Selection Ratio | | | |
|---|---|---|---|---|---|---|
| | | | 0.2 | 0.4 | 0.6 | 0.8 |
| SOP+ | CIFAR-10N (Worst) | MaxProb | 82.7 | 88.1 | 91.3 | 92.5 |
| | | DiffProb | 82.5 | 88.5 | 91.2 | 92.5 |
| | CIFAR-100N | MaxProb | 50.2 | 59.1 | 63.9 | 65.7 |
| | | DiffProb | 49.2 | 59.3 | 64.1 | 66.0 |

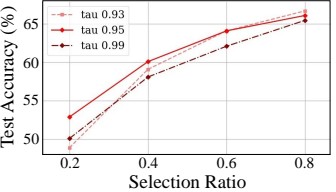

Figure 4: Effect of the neighborhood threshold $\tau$ on Prune4ReL$_B$.

## 4.3 Necessity of Data Pruning with Re-labeling under Label Noise

Table 2 shows the superiority of the Re-labeling models over the standard learning model, *i.e.*, only with the cross-entropy loss, for data pruning under label noise. When combined with the data pruning methods, the performance of the Re-labeling models such as DivideMix and SOP+ significantly surpasses those of the standard models on CIFAR-10N and CIFAR-100N by up to 21.6%. That is, the re-labeling capacity can be well preserved by a proper data pruning strategy. This result demonstrates the necessity of data pruning for the Re-labeling models in the presence of noisy labels.

## 4.4 Ablation Studies

**Effect of Confidence Metrics.** Prune4ReL can be integrated with various metrics for the confidence of predictions in Eq. (6). In our investigation, we consider two widely-used metrics: (1) MaxProb, which represents the maximum value of softmax probability and (2) DiffProb, which measures the difference between the highest and second highest softmax probabilities. Table 3 shows the effect of the two confidence metrics on the test accuracy of SOP+ on CIFAR-10N (Worst) and CIFAR-100N. The result indicates that both metrics perform similarly and yield higher accuracy compared to existing data pruning baselines, which demonstrates Prune4ReL is open to the choice of the confidence metric.

**Effect of Neighborhood Size.** Prune4ReL involves a hyperparameter $\tau$ in Eq. (6) to determine the neighborhood of each example, where a larger (smaller) value introduces a fewer (more) number of neighbors. Figure 4 shows the effect of $\tau \in \{0.93, 0.95, 0.99\}$ on the test accuracy of SOP+ trained on CIFAR-100N with varying selection ratios. In general, Prune4ReL shows better or comparable performance than other sample selection baselines. Among them, Prune4ReL with $\tau = 0.95$ shows a satisfactory accuracy across varying selection ratios. With a smaller value of $\tau = 0.93$, it shows the best performance in a high selection ratio of 0.8, but it becomes less effective in low selection ratios, due to the increasing influence of noisy examples caused by a relatively large neighborhood size. On the contrary, with a large value of $\tau = 0.99$, it primarily selects clean yet easy examples due to a small neighborhood size, leading to a relatively less improvement in performance.

## 4.5 In-depth Analysis of Noisy Examples in Selected Subset

**Noise Ratio of Selected Subset.** Table 4 shows the ratio of noisy examples in the subset selected by each sample selection method. SmallLoss shows a very low ratio of noisy examples in the subset because it prefers clean examples. Many data pruning methods, including Margin, GraNd, SSP, and Moderate (for CIFAR-100N), tend to select a higher ratio of noisy examples compared with that of each original dataset since they prefer to select hard examples. On the other hand, Prune4ReL selects a low ratio of noisy examples when the subset size is small and gradually increases the noise ratio as the subset size increases. This result indicates that Prune4ReL expands the confident subset through Algorithm 1—*i.e.*, selecting the most confident (clean) examples first and then trying to select less confident (hard or noisy) neighbors to ensure accurate re-labeling. While some baselines, such as kCenterGreedy, Forget, and Moderate (for CIFAR-10N), also select a somewhat low ratio of noisy examples, their data pruning performances are worse than Prune4ReL because the quality (or self-correctability) of noisy examples is not considered when selecting the subset, which is further investigated in Table 5.

**Self-correctability of Selected Noisy Examples.** Table 5 shows the self-correctability of selected subsets, which indicates the ratio of correctly re-labeled noisy examples out of all selected noisy

Table 4: Ratio (%) of noisy examples in the selected subset.

| Re-label Model | Selection Methods | CIFAR-10N (Random, ≈18%) | | | | CIFAR-10N (Worst, ≈40%) | | | | CIFAR-100N (Noisy, ≈40%) | | | |
|---|---|---|---|---|---|---|---|---|---|---|---|---|---|
| | | 0.2 | 0.4 | 0.6 | 0.8 | 0.2 | 0.4 | 0.6 | 0.8 | 0.2 | 0.4 | 0.6 | 0.8 |
| SOP+ | SmallL | 0.1 | 0.2 | 1.0 | 4.0 | 0.8 | 3.6 | 11.5 | 26.2 | 3.5 | 8.7 | 16.3 | 27.4 |
| | Margin | 29.7 | 25.7 | 22.5 | 19.7 | 54.6 | 52.1 | 48.5 | 44.6 | 61.5 | 56.8 | 51.6 | 46.2 |
| | kCenter | 19.0 | 18.8 | 19.1 | 18.6 | 40.0 | 41.1 | 41.6 | 42.1 | 37.5 | 38.7 | 39.9 | 40.4 |
| | Forget | 17.0 | 17.7 | 17.5 | 17.8 | 37.7 | 38.8 | 39.2 | 40.2 | 37.9 | 34.6 | 33.0 | 36.8 |
| | GraNd | 67.5 | 41.7 | 28.6 | 21.5 | 91.5 | 79.4 | 64.2 | 50.1 | 93.9 | 57.3 | 61.2 | 49.3 |
| | SSP | 25.2 | 23.7 | 21.6 | 19.5 | 48.5 | 46.4 | 43.2 | 42.1 | 52.7 | 52.3 | 46.8 | 43.0 |
| | Moderate | 6.6 | 7.1 | 8.4 | 13.5 | 31.7 | 33.4 | 34.7 | 40.0 | 33.2 | 54.6 | 60.2 | 64.6 |
| | **Pr4ReL** | 17.0 | 18.7 | 19.3 | 22.5 | 38.7 | 42.7 | 43.5 | 46.5 | 28.3 | 29.1 | 33.3 | 37.2 |

Table 5: Ratio of correctly re-labeled noisy examples in the selected subset (denoted as % *Correct*).

| Re-label Model | Selection Methods | CIFAR-10N (Random) | | |
|---|---|---|---|---|
| | | *Test Acc.* | *% Noisy* | *% Correct* |
| SOP+ | kCenter | 86.3 | 19.0 | 75.2 |
| | Forget | 82.4 | 17.0 | 61.7 |
| | **Pr4ReL** | 88.1 | 17.0 | **90.3** |

Table 6: Data pruning performance on ImageNet with a 20% synthetic label noise.

| Re-label Model | Selection Methods | ImageNet-1K (Syn, ≈20%) | | | |
|---|---|---|---|---|---|
| | | 0.05 | 0.1 | 0.2 | 0.4 |
| SOP+ | Uniform | 27.8 | 42.5 | 52.7 | 59.2 |
| | SmallL | 22.8 | 31.4 | 42.7 | 54.4 |
| | Forget | 4.1 | 8.3 | 50.6 | 57.2 |
| | **Pr4ReL$_B$** | **30.2** | **44.3** | **53.5** | **60.0** |

examples. Here, we compare Prune4ReL with kCenterGreedy and Forgetting on CIFAR-10N (Random) with the selection ratio of 0.2, where the ratio of noisy examples (*i.e.*, $\%Noisy$) in the selected subset of each method is similar (*i.e.*, from 17% to 19%). Although these methods select almost equal amounts of noisy examples, there were differences in the self-correctability (*i.e.*, $\%Correct$) of the selected subsets. Noisy examples selected by Prune4ReL are mostly self-correctable as it maximizes the total neighborhood confidence of the training set. In contrast, those selected by existing data pruning methods such as kCenter and Forget are not guaranteed to be self-correctable. This result confirms that Prune4ReL not only selects a low ratio of noisy examples but also considers the quality of the selected subset in terms of maximizing re-labeling accuracy. Therefore, Prune4ReL fully takes advantage of the Re-labeling methods.

## 4.6 Results on ImageNet-N with Synthetic Label Noise

We further validate the efficacy of Prune4ReL on ImageNet-N by injecting synthetic label noise of 20% to the commonly-used benchmark dataset ImageNet-1K (see Appendix D for details). Table 6 shows the test accuracy of Prune4ReL and three representative sample selection baselines with varying selection ratios of $\{0.05, 0.1, 0.2, 0.4\}$. Similar to the result in Section 4.2, Prune4ReL consistently outperforms the baselines by up to 8.6%, thereby adding more evidence of the superiority of Prune4ReL. In addition, owing to its great computation efficiency, Prune4ReL is able to scale to ImageNet-N, a large-scale dataset with approximately 1.2M training examples.

## 5 Conclusion

We present a noise-robust data pruning method for Re-labeling, called Prune4ReL, that finds a subset that maximizes the total neighborhood confidence of the training examples, thereby maximizing the re-labeling accuracy and generalization performance. To identify a subset that maximizes the re-labeling accuracy, Prune4ReL introduces a novel metric, the *reduced neighborhood confidence*, which is the prediction confidence of each neighbor example in the selected subset, and the effectiveness of this metric in estimating the re-labeling capacity of a subset is theoretically and empirically validated. Furthermore, we optimize Prune4ReL with an efficient greedy algorithm that expands the subset by selecting the example that contributes the most to increasing the total reduced neighborhood confidence. Experimental results demonstrate the substantial superiority of Prune4ReL compared to existing data pruning methods in the presence of label noise.

## Acknowledgement

This work was supported by Institute of Information & Communications Technology Planning & Evaluation (IITP) grant funded by the Korea government (MSIT) (No. 2020-0-00862, DB4DL: High-Usability and Performance In-Memory Distributed DBMS for Deep Learning and No. 2022-0-00157, Robust, Fair, Extensible Data-Centric Continual Learning).

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
