# Robust Data Pruning under Label Noise via Maximizing Re-labeling Accuracy
# (Supplementary Material)

## A   Complete Proof of Theorem 3.4

The $\alpha$-expansion and $\beta$-separation assumptions hold for the training set $\tilde{\mathcal{D}}$. Then, following the re-labeling theory [45], minimizing the self-consistency loss forces the classifier into correcting the erroneous labels and improving the training accuracy, as presented in Lemma A.1.

**Lemma A.1.**  (RE-LABELING BOUND). *Suppose $\alpha$-expansion and $\beta$-separation assumptions hold for the training set $\tilde{\mathcal{D}}$. Then, for a Re-labeling minimizer $\theta_{\tilde{\mathcal{D}}}$ on $\tilde{\mathcal{D}}$, we have*

$$Err(\theta_{\tilde{\mathcal{D}}}) \leq \frac{2 \cdot Err(\theta_{\mathcal{M}})}{\alpha - 1} + \frac{2 \cdot \alpha}{\alpha - 1} \cdot \beta, \tag{8}$$

*where $Err(\cdot)$ is a training error on ground-truth labels, and $\theta_{\mathcal{M}}$ is a model trained with the supervised loss in Eq.* (1) *on a minimum (or given) clean set $\mathcal{M} \subset \mathcal{S}$.*

*Proof.*  Refer to [45] for the detailed concept and proof.  □

This lemma is used for proving Theorem 3.4. Since $\alpha_{\mathcal{S}}$ indicates the average number of augmentation neighbors in $\mathcal{S}$, we can transform Eq. (8) using $\alpha_{\mathcal{S}}$,

$$Err(\theta_{\mathcal{S}}) \leq \frac{2 \cdot Err(\theta_{\mathcal{M}})}{(1/|\mathcal{S}|) \sum_{x \in \mathcal{S}} \mathbb{1}_{[x' \in \mathcal{N}(x)]}} + \frac{2 \cdot \alpha_{\mathcal{S}}}{\alpha_{\mathcal{S}} - 1} \cdot \beta_{\mathcal{S}}. \tag{9}$$

Assume that the training error of the minimum clean set $\mathcal{M}$ in the selected subset $\mathcal{S}$ is proportional to the inverse of the confidence of $x \in \mathcal{S}$, since the performance of the standard learner is often correlated to the confidence of training examples. Then, Eq. (9) becomes

$$\begin{aligned} Err(\theta_{\mathcal{S}}) &\leq \frac{2 \cdot |\mathcal{S}| \cdot Err(\theta_{\mathcal{M}})}{\sum_{x \in \mathcal{S}} C(x) \sum_{x \in \mathcal{S}} \mathbb{1}_{[x' \in \mathcal{N}(x)]}} + \frac{2 \cdot \alpha_{\mathcal{S}}}{\alpha_{\mathcal{S}} - 1} \cdot \beta_{\mathcal{S}} \\ &\leq \frac{2 \cdot |\mathcal{S}| \cdot Err(\theta_{\mathcal{M}})}{\sum_{x \in \mathcal{S}} \mathbb{1}_{[x' \in \mathcal{N}(x)]} C(x)} + \frac{2 \cdot \alpha_{\mathcal{S}}}{\alpha_{\mathcal{S}} - 1} \cdot \beta_{\mathcal{S}}, \end{aligned} \tag{10}$$

where the last inequality holds because of Hölder's inequality with two sequence variables. Therefore, $Err(\theta_{\mathcal{S}}) \leq \frac{2 \cdot |\mathcal{S}| \cdot Err(\theta_{\mathcal{M}})}{\sum_{x \in \tilde{\mathcal{S}}} C_{\mathcal{N}}(x; \mathcal{S})} + \frac{2 \cdot \alpha_{\mathcal{S}}}{\alpha_{\mathcal{S}} - 1} \cdot \beta_{\mathcal{S}}$, and this concludes the proof of Theorem 3.4.

## B   Details for Prune4ReL$_B$

---

**Algorithm 2** Greedy Balanced Neighborhood Confidence (Prune4ReL$_B$)

---

INPUT:  $\tilde{\mathcal{D}}$: training set, $\tilde{\mathcal{D}}_j(\subset \tilde{\mathcal{D}})$: set of training examples with a $j$-th class, $s$: target subset size, and $C(x)$: confidence of $x$ calculated from a warm-up classifier

1:  Initialize $\mathcal{S} \leftarrow \emptyset; \forall x \in \tilde{\mathcal{D}}, \hat{C}_{\mathcal{N}}(x) = 0$
2:  **while** $|\mathcal{S}| < s$ **do**
3:     **for** $j = 1$ to $c$ **do**
4:        $x = \operatorname{argmax}_{x \in \tilde{\mathcal{D}}_j \setminus \mathcal{S}} \ \boldsymbol{\sigma}(\hat{C}_{\mathcal{N}}(x) + C(x)) - \boldsymbol{\sigma}(\hat{C}_{\mathcal{N}}(x))$
5:        $\mathcal{S} = \mathcal{S} \cup \{x\}$
6:        **for all** $v \in \tilde{\mathcal{D}}$ **do**
7:           $\hat{C}_{\mathcal{N}}(v) \mathrel{+}= \mathbb{1}_{[sim(x,v) \geq \tau]} \cdot sim(x, v) \cdot C(x)$
8:        **if** $|\mathcal{S}| = s$ **do**
9:           **return** $\mathcal{S}$
10: **end**
OUTPUT:  Final selected subset $\mathcal{S}$

---

Algorithm 2 describes the class-balanced version of our greedy algorithm. We first divide the entire training set into $c$ groups according to the noisy label of each example, under the assumption that the number of correctly labeled examples is much larger than that of incorrectly labeled examples in practice [7]. Similar to Algorithm 1 in Section 3.2, we begin with an empty set $\mathcal{S}$ and initialize the reduced neighborhood confidence $\hat{C}_{\mathcal{N}}$ to 0 for each training example (Line 1). Then, by iterating class $j$, we select an example $x$ that maximizes the marginal benefit $\boldsymbol{\sigma}(\hat{C}_{\mathcal{N}}(x)+C(x))-\boldsymbol{\sigma}(\hat{C}_{\mathcal{N}}(x))$ within the set $\tilde{\mathcal{D}}_j(\subset \tilde{\mathcal{D}})$ and add it to the selected subset $\mathcal{S}$ (Lines 3–5). Next, we update the reduced neighborhood confidence $\hat{C}_{\mathcal{N}}$ of each example in the entire training set by using the confidence and the similarity score to the selected example $x$ (Lines 6–7). We repeat this procedure until the size of the selected subset $\mathcal{S}$ meets the target size $s$ (Lines 8–9).

## C    Complete Proof of Theorem 3.5

We complete Theorem 3.5 by proving the *monotonicity* and *submodularity* of Eq. (6) in Lemmas C.1 and C.2, under the widely proven fact that the monotonicity and submodularity of a combinatorial objective guarantee the greedy selection to get an objective value within $(1-1/e)$ of the optimum [54].

**Lemma C.1.** (MONOTONICITY). *Our data pruning objective in Eq.* (6)*, denoted as $OBJ$, is monotonic. Formally,*

$$\forall\, \mathcal{S} \subset \mathcal{S}', \ \ OBJ(\mathcal{S}) \leq OBJ(\mathcal{S}'). \tag{11}$$

*Proof.*

$$OBJ(\mathcal{S}') = \sum_{x_i \in \tilde{\mathcal{D}}} \boldsymbol{\sigma}\big(\hat{C}_{\mathcal{N}}(x_i; \mathcal{S}')\big) = \sum_{x_i \in \tilde{\mathcal{D}}} \boldsymbol{\sigma}\big( \sum_{x_j \in \mathcal{S}'} \mathbb{1}_{[sim(x_i,x_j)\geq \tau]} \cdot sim\big(x_i, x_j\big) \cdot C(x_j) \ \big)$$

$$= \sum_{x_i \in \tilde{\mathcal{D}}} \boldsymbol{\sigma}\big( \sum_{x_j \in \mathcal{S}} \mathbb{1}_{[sim(x_i,x_j)\geq \tau]} \cdot sim\big(x_i, x_j\big) \cdot C(x_j) + \sum_{x_j \in \mathcal{S}'\backslash \mathcal{S}} \mathbb{1}_{[sim(x_i,x_j)\geq \tau]} \cdot sim\big(x_i, x_j\big) \cdot C(x_j)\big) \tag{12}$$

$$\geq \sum_{x_i \in \tilde{\mathcal{D}}} \boldsymbol{\sigma}\big( \sum_{x_j \in \mathcal{S}} \mathbb{1}_{[sim(x_i,x_j)\geq \tau]} \cdot sim\big(x_i, x_j\big) \cdot C(x_j)\big) = \sum_{x_i \in \tilde{\mathcal{D}}} \boldsymbol{\sigma}\big(\hat{C}_{\mathcal{N}}(x_i; \mathcal{S})\big) = OBJ(\mathcal{S}),$$

where the inequality holds because of the non-decreasing property of the utility function $\sigma$. Therefore, $OBJ(\mathcal{S}) \leq OBJ(\mathcal{S}')$. $\qquad\square$

**Lemma C.2.** (SUBMODULARITY). *Our objective in Eq.* (6) *is submodular. Formally,*

$$\forall\, \mathcal{S} \subset \mathcal{S}' \text{ and } \forall x \notin \mathcal{S}', \ \ OBJ(\mathcal{S} \cup \{x\}) - OBJ(\mathcal{S}) \geq OBJ(\mathcal{S}' \cup \{x\}) - OBJ(\mathcal{S}'). \tag{13}$$

*Proof.* For notational simplicity, let $x_i$ be $i$, $x_j$ be $j$, and $\mathbb{1}_{[sim(x_i,x_j)\geq \tau]} \cdot sim\big(x_i, x_j\big) \cdot C(x_j)$ be $C_{ij}$. Then, Eq. (13) can be represented as

$$\sum_{i\in\tilde{\mathcal{D}}} \boldsymbol{\sigma}\big( \sum_{j\in\mathcal{S}} C_{ij} + C_{ix}\big) - \sum_{i\in\tilde{\mathcal{D}}} \boldsymbol{\sigma}\big( \sum_{j\in\mathcal{S}} C_{ij}\big) \geq \sum_{i\in\tilde{\mathcal{D}}} \boldsymbol{\sigma}\big( \sum_{j\in\mathcal{S}'} C_{ij} + C_{ix}\big) - \sum_{i\in\tilde{\mathcal{D}}} \boldsymbol{\sigma}\big( \sum_{j\in\mathcal{S}'} C_{ij}\big). \tag{14}$$

Proving Eq. (14) is equivalent to proving the decomposed inequality for each example $x_i \in \tilde{\mathcal{D}}$,

$$\boldsymbol{\sigma}\big( \sum_{j\in\mathcal{S}} C_{ij} + C_{ix}\big) - \boldsymbol{\sigma}\big( \sum_{j\in\mathcal{S}} C_{ij}\big) \geq \boldsymbol{\sigma}\big( \sum_{j\in\mathcal{S}'} C_{ij} + C_{ix}\big) - \boldsymbol{\sigma}\big( \sum_{j\in\mathcal{S}'} C_{ij}\big)$$

$$= \boldsymbol{\sigma}\big( \sum_{j\in\mathcal{S}} C_{ij} + \sum_{j\in\mathcal{S}'\backslash\mathcal{S}} C_{ij} + C_{ix}\big) - \boldsymbol{\sigma}\big( \sum_{j\in\mathcal{S}} C_{ij} + \sum_{j\in\mathcal{S}'\backslash\mathcal{S}} C_{ij}\big). \tag{15}$$

Since $\mathcal{S}$, $\mathcal{S}'\backslash\mathcal{S}$, and $\{x\}$ do not intersect each other, we can further simplify Eq. (15) with independent scala variables such that

$$\boldsymbol{\sigma}\big(a + \epsilon\big) - \boldsymbol{\sigma}\big(a\big) \geq \boldsymbol{\sigma}\big(a + b + \epsilon\big) - \boldsymbol{\sigma}\big(a + b\big), \tag{16}$$

where $a = \sum_{j\in\mathcal{S}} C_{ij}$, $b = \sum_{j\in\mathcal{S}'\backslash\mathcal{S}} C_{ij}$, and $\epsilon = C_{ix}$.

Since the utility function $\sigma$ is *concave*, by the definition of concavity,

$$\frac{\boldsymbol{\sigma}\big(a + \epsilon\big) - \boldsymbol{\sigma}\big(a\big)}{(a + \epsilon - a)} \geq \frac{\boldsymbol{\sigma}\big(a + b + \epsilon\big) - \boldsymbol{\sigma}\big(a + b\big)}{(a + b + \epsilon - (a + b))}. \tag{17}$$

The denominators of both sides of the inequality become $\epsilon$, and Eq. (17) can be transformed to Eq. (16). Therefore, Eq. (16) should hold, and $OBJ(\mathcal{S}\cup\{x\})-OBJ(\mathcal{S}) \geq OBJ(\mathcal{S}'\cup\{x\})-OBJ(\mathcal{S}')$. $\quad\square$

Table 7: Summary of the hyperparameters for training SOP+ and DivideMix on the CIFAR-10N/100N, Webvision, and Clothing-1M datasets.

| | Hyperparamters | CIFAR-10N | CIFAR-100N | WebVision | Clothing-1M |
|---|---|---|---|---|---|
| **Training Configuration** | architecture | PreAct PresNet18 | PreAct PresNet18 | InceptionResNetV2 | ResNet-50 (pretrained) |
| | warm-up epoch | 10 | 30 | 10 | 0 |
| | training epoch | 300 | 300 | 100 | 10 |
| | batch size | 128 | 128 | 32 | 32 |
| | learning rate (lr) | 0.02 | 0.02 | 0.02 | 0.002 |
| | lr scheduler | Cosine Annealing | Cosine Annealing | MultiStep-50th | MultiStep-5th |
| | weight decay | $5 \times 10^{-4}$ | $5 \times 10^{-4}$ | $5 \times 10^{-4}$ | 0.001 |
| **DivideMix** | $\lambda_U$ | 1 | 1 | | 0.1 |
| | $\kappa$ | 0.5 | 0.5 | | 0.5 |
| | $T$ | 0.5 | 0.5 | – | 0.5 |
| | $\gamma$ | 4 | 4 | | 0.5 |
| | $M$ | 2 | 2 | | 2 |
| **SOP+** | $\lambda_C$ | 0.9 | 0.9 | 0.1 | |
| | $\lambda_B$ | 0.1 | 0.1 | 0 | |
| | lr for $u$ | 10 | 1 | 0.1 | – |
| | lr for $v$ | 100 | 100 | 1 | |

By Lemmas C.1 and C.2, the monotonicity and submodularity of Eq. (6) hold. Therefore, Eq. (7) naturally holds, and this concludes the proof of Theorem 3.5.

# D    Details for Constructing ImageNet-N

Since ImageNet-1K is a clean dataset with no known real label noise, we inject the synthetic label noise to construct ImageNet-N. Specifically, we inject *asymmetric* label noise to mimic real-world label noise following the prior noisy label literature [10]. When a target noise ratio of ImageNet-N is $r\%$, we randomly select $r\%$ of the training examples for each class $c$ in ImageNet-1K and then flip their label into class $c + 1$, *i.e.*, class 0 into class 1, class 1 into class 2, and so on. This flipping is reasonable because consecutive classes likely belong to the same high-level category. For the selected examples with the last class 1000, we flip their label into class 0.

# E    Implementation Details

Table 7 summarizes the overall training configurations and hyperparameters used to train the two Re-labeling models, DivideMix and SOP+. The hyperparameters for DivideMix and SOP+ are favorably configured following the original papers. DivideMix [13] has multiple hyperparameters: $\lambda_U$ for weighting the self-consistency loss, $\kappa$ for selecting confidence examples, $T$ for sharpening prediction probabilities, $\gamma$ for controlling the Beta distribution, and $M$ for the number of augmentations. For both CIFAR-10N and CIFAR-100N, we use $\lambda_U = 1$, $\kappa = 0.5$, $T = 0.5$, $\gamma = 4$, and $M = 2$. For Clothing-1M, we use $\lambda_U = 0.1$, $\kappa = 0.5$, $T = 0.5$, $\gamma = 0.5$, and $M = 2$. SOP+ [33] also involves several hyperparameters: $\lambda_C$ for weighting the self-consistency loss, $\lambda_B$ for weighting the class-balance, and learning rates for training its additional variables $u$ and $v$. For CIFAR-10N, we use $\lambda_C = 0.9$ and $\lambda_B = 0.1$, and set the learning rates of $u$ and $v$ to 10 and 100, respectively. For CIFAR-100N, we use $\lambda_C = 0.9$ and $\lambda_B = 0.1$, and set the learning rates of $u$ and $v$ to 1 and 100, respectively. For WebVision, we use $\lambda_C = 0.1$ and $\lambda_B = 0$, and set the learning rates of $u$ and $v$ to 0.1 and 1, respectively.

Besides, the hyperparameters for all data pruning algorithms are also favorably configured following the original papers. For Forgetting [14], we calculate the forgetting event of each example throughout the warm-up training epochs in each dataset. For GraNd [15], we train ten different warm-up classifiers and calculate the per-sample average of the norms of the gradient vectors obtained from the ten classifiers.

# F    Limitation and Potential Negative Societal Impact

**Limitation.** Although Prune4ReL has demonstrated consistent effectiveness in the classification task with real and synthetic label noises, we have not validated its applicability on datasets with open-set noise or out-of-distribution examples [55, 56]. Also, we have not validated its applicability to state-of-the-art deep learning models, such as large language models [3] and vision-language models [4]. This verification would be valuable because the need for data pruning in the face of annotation noise is consistently high across a wide range of real-world tasks. In addition, Prune4ReL has not been validated in other realistic applications of data pruning, such as continual learning [57] and neural architecture search [58]. In these scenarios, selecting informative examples is very important, and we leave them for future research.

**Potential Negative Societal Impact.** We consider how to preserve the model performance while reducing the computation costs, which can even reduce substantial energy consumption, *e.g.*, $CO_2$ emission. Hence, it is hard to apply to any negative applications, and there is no discussion of potential negative social impact.