# OpenReview forum: "Robust Data Pruning under Label Noise via Maximizing Re-labeling Accuracy"
_NeurIPS.cc/2023/Conference — NeurIPS 2023 poster_

### Official Review · Reviewer_rFfD · 2023-06-15

**Soundness:** 3 good
**Presentation:** 3 good
**Contribution:** 3 good
**Rating:** 6
**Confidence:** 4

**Summary:**

This paper studies data pruning in noisy label setting. Built on re-labeling models, it proposes Prune4ReL that finds a subset to maximize the re-labeling accuracy. In particular, it introduces neighborhood confidence as the criteria for selection, as well as a greedy algorithm to select the subset. Evaluation show its superiority over prior data pruning methods.

**Strengths:**

The target setting is interesting and well motivated. The proposed neighborhood confidence and Prune4ReL is sound with theoretical guarantee. In addition, the evaluations are comprehensive and experimental results look promising.

**Weaknesses:**

I don't see any major weakness, but I do have some comments:

1. The proposed data pruning method does require model training as many sample selection methods for robust learning do; I think these methods should also be considered as baselines and compared, even though they are not specifically designed for data pruning. Currently, the authors only compare SmallLoss, which is quite outdated.

2. There are some related work in robust learning field that also leverage neighborhood information, but the authors did not include and discuss them, eg, "Learning with Neighbor Consistency for Noisy Labels"

**Questions:**

See weakness above

**Limitations:**

The authors clearly stated the limitations in appendix.

---

> ### Author Rebuttal · Authors · 2023-08-10
>
> `Q1. The proposed data pruning method does require model training as many sample selection methods for robust learning do; I think these methods should also be considered as baselines and compared, even though they are not specifically designed for data pruning.`
>
> Thank you very much for helping us improve our paper. Many sample selection methods, including the small-loss trick, have been developed for the purpose of removing noisy examples from the training set. However, these methods do not consider the importance of examples in terms of re-labeling. Thus, a new data pruning method should ***not*** completely remove noisy examples but select them if they are expected to be relabeled correctly. We expect that other sample selection methods would also perform worse than ours because they do not try to select noisy examples which could be potentially helpful. According to your suggestion, we will try to add more decent sample selection baselines such as Co-teaching+[a] in the final version.
>
> [a] Xingrui Yu, Bo Han, Jiangchao Yao, Gang Niu, Ivor W. Tsang, Masashi Sugiyama: How does Disagreement Help Generalization against Label Corruption? ICML 2019: 7164-7173
>
>
> `Q2. There are some related work in robust learning field that also leverage neighborhood information, but the authors did not include and discuss them, e.g., "Learning with Neighbor Consistency for Noisy Labels"`
>
> Thank you for bringing up an important topic for discussion. The method NCR[b] that you mentioned utilizes consistency regularization among neighbors so that examples with similar feature representations produce similar outputs. Consequently, NCR and Prune4Rel share the phiolosophy that neighbor examples are useful in the presence of label noise. Nevertheless, the objectives of the two methods to utilizing the neighborhood are distinct: NCR aims to reduce the impact of incorrect labels, whereas Prune4Rel seeks to determine the contribution of an example to the re-labeling accuracy. The final version will include this discussion.
>
> [b] Ahmet Iscen, Jack Valmadre, Anurag Arnab, Cordelia Schmid: Learning with Neighbor Consistency for Noisy Labels. CVPR 2022: 4662-4671

---

> > ### Comment · Reviewer_rFfD · 2023-08-10
> > **Post rebuttal**
> >
> > I read the author's rebuttal and would keep my score. In addition, I think this work is also related but missed in discussion/comparison: Resolving Training Biases via Influence-based Data Relabeling.

---

> > > ### Author Response · Authors · 2023-08-11
> > > **Thank you**
> > >
> > > Thank you very much for suggesting an additional reference.  We will definitely include it for discussion or comparison purposes in the final version.

---

### Official Review · Reviewer_ALmL · 2023-06-25

**Soundness:** 2 fair
**Presentation:** 3 good
**Contribution:** 2 fair
**Rating:** 5
**Confidence:** 4

**Summary:**

This paper studies the task of data pruning, specifically in the setting of label noise. The authors propose a method to perform data pruning by maximizes the total neighborhood confidence of the training examples (which is equivalent to maximizing the relabeling accuracy).

The authors theoretically analyze this particular setting, bounding the error of a model trained on the subset that satisfies the expansion and separation assumptions from prior work. Their resulting bound is inversely proportional to the neighborhood confidence.

Their empirical results show mixed results in the comparison against existing methods; namely, in Table 1, the results are slightly better than existing baselines on a small number of tasks but predominantly match the performance of existing methods. In table 2, it also seems that the existing method Forget and kCenter seem to select similar or fewer noisy examples than the proposed method (except in the case of CIFAR-100N with 0.2 or 0.4).


**Strengths:**

The authors present a theorem that shows that the error of a model trained on a subset of data is inversely related to the neighborhood confidence. They propose a method that maximizes the neighborhood confidence, which in turn minimizes this bound.

The authors show in Figure 3 that the test accuracy with SOP increases when using a subset produced by the proposed method. However, it is unclear why this is necessarily the case, since there are similar or fewer noisy examples selected by the Forget baseline method.

The authors provide ablation studies that study the effect of neighborhood threshold $\tau$ and with different confidence metrics.


**Weaknesses:**

The bolding strategies in Table 1 are a bit misleading; in many scenarios, a baseline achieves the same performance as the proposed method (and sometimes achieves a smaller variance), but the proposed method is still listed in bold. In fact, the reported method achieves the same performance across many of the tasks.

Similarly, the results in Table 2 show that the Forget baseline seems to select subsets with a smaller ratio of noisy examples when compared to the proposed method. However, in subsection 4.3, the authors claim that this method selects a high ratio of noisy examples.


**Questions:**

N/A

**Limitations:**

The authors have adequately addressed the limitations.

---

> ### Author Rebuttal · Authors · 2023-08-10
>
> `Q1. Their empirical results show mixed results in the comparison against existing methods. In Table 1, the results are slightly better than existing baselines on a small number of tasks but predominantly match the performance of existing methods`
>
> Thank you very much for your careful review. We acknowledge your concerns but respectfully argue that **our empirical results are promising** in three ways.
>
> > **(1) Prune4Rel *consistently* outperforms in various datasets and selection ratios.** No baseline consistently shows comparable accuracy in Table 1. To illustrate our claim, the average over different selection ratios for each dataset with SOP+ is calculated as follows.
>
> | Methods        | CIFAR-10N Random | CIFAR-10N Worst | CIFAR-100N Noisy |
> | -------------- | -------- | -------- | -------- |
> | Uniform        | 91.8     | 88.0     | 56.9     |
> | SmallL         | 87.2     | 86.1     | 59.6     |
> | Margin         | 79.9     | 71.2     | 42.0     |
> | kCenter        | 92.0     | 88.4     | 56.9     |
> | Forget         | 91.2     | 85.4     | 55.6     |
> | GraNd          | 64.2     | 44.7     | 32.7     |
> | **Pr4Rel**     | 92.7     | 89.2     | 59.5     |
> | **Pr4Rel$_B$** | **92.8** | **89.3** | **60.8** |
>
> > **(2) The improvement becomes more pronounced when the selection ratio is *relatively low* (e.g., 20% in Table 1).** It is relatively hard to show the difference at high selection ratios, because not many examples are pruned. Table 1 shows that slightly better or comparable accuracy tends to appear at high selection ratios.
>
> > **(3) The results on another dataset reaffirm the superiority of Prune4Rel.** The table in *Q2 of the reviewer jBWT* is obtained for ImageNet-1K with synthetic flip noises.
>
> Overall, we hope that the reviewer's concerns on empirical results are resolved by our three supporting arguments. We will enrich Table 1 by adding the averages and new results in the final version.
>
>
> `Q2. In Table 2, it seems that the existing method Forget and kCenter seem to select similar or fewer noisy examples than the proposed method (except in the case of CIFAR-100N with 0.2 or 0.4).`
>
> The goodness of a selected subset cannot be determined based solely on the proportion of noisy examples; the **quality** (or **self-correctability**) of these noisy examples must also be considered. Noisy examples selected by Prune4Rel are mostly self-correctable because it maximizes the total neighborhood confidence of the training set. In contrast, those selected by existing data pruning methods such as kCenter and Forget are ***not*** guaranteed to be self-correctable. To contrast the proportion of self-correctable examples in kCenter, Forget, and Prune4Rel, we plan to expand Table 2 as follows. Here is the proportion of self-correctable examples among the noisy ones within a selected subset at a selection ratio of 0.2.
>
> | Model | Methods    | CIFAR-10N Random |  |  |
> | ----- | ---------- | -------- | -------- | -------- |
> |       |            | Test Accuracy | % Noisy | % (Self-Correct / Noisy) |
> | SOP+  | kCenter    | 86.3     | 19.0 | 75.2     |
> |       | Forget     | 82.4     | 17.0 | 61.7     |
> |       | **Pr4Rel** | 88.2     | 17.0 | **90.3** |
>
> Therefore, the noisy examples selected by kCenter and Forget may harm the generalizability of a model, whereas those selected by Prune4Rel are rather useful for training.
>
> In Section 4.3, we meant to say that Prune4Rel selects more high-quality noisy examples aggressively as the subset size increases, based on the increased confidence. We recognize that the current writing is not very clear, and we will revise it accordingly.
>
>
> `Q3. The authors show in Figure 3 that the test accuracy with SOP+ increases when using a subset produced by the proposed method. However, it is unclear why this is necessarily the case, since there are similar or fewer noisy examples selected by the Forget baseline method.`
>
> This question is answered by the **response to Q2**. The contribution of noisy examples to a model's relabeling capability is low in kCenter and Forget, whereas it is very high in Prune4Rel, despite the fact that the proportion of noisy examples is comparable.
>
>
> `Q4. The bolding strategies in Table 1 are a bit misleading; in many scenarios, a baseline achieves the same performance as the proposed method (and sometimes achieves a smaller variance), but the proposed method is still listed in bold.`
>
> We apologize for any confusion that our current presentation may have caused. We intended to highlight only our methods, unless they were inferior to any baseline. We will revise the strategy for bolding such that the highest values (including ties) are highlighted. For example, part of Table 1 for SOP+ will be changed to the following.
>
> | Methods        |         | CIFAR-10N | Random   |          |         | CIFAR-10N | Worst    |          |        | CIFAR-100N | Noisy    |          |
> | -------------- | -------- | -------- | -------- | -------- | -------- | -------- | -------- | -------- | -------- | -------- | -------- | -------- |
> |                | 0.2      | 0.4      | 0.6      | 0.8      | 0.2      | 0.4      | 0.6      | 0.8      | 0.2      | 0.4      | 0.6      | 0.8      |
> | SmallL         | 77.6     | 86.2     | 90.7     | 94.3     | 78.8     | 84.1     | 89.3     | 92.3     | 48.5     | 59.8     | 63.9     | **66.1** |
> | Margin         | 52.1     | 79.6     | 92.6     | 95.1     | 45.7     | 61.8     | 84.6     | **92.5** | 20.0     | 34.4     | 50.4     | 63.3     |
> | kCenter        | 86.3     | 92.2     | 94.1     | **95.3** | 81.9     | 88.0     | 91.3     | 92.3     | 44.8     | 55.9     | 61.6     | 65.2     |
> | **Pr4Rel**     | 88.2     | 93.0     | **94.4** | 95.1     | 83.4     | **89.3** | **91.5** | **92.5** | 49.0     | 59.1 | **64.1** | 65.7 |
> | **Pr4Rel$_B$** | **88.6** | **93.1** | 94.2     | **95.3** | **84.2** | 89.1     | 91.3     | **92.5** | **52.9** | **60.1** | **64.1** | **66.1** |

---

> > ### Comment · Reviewer_ALmL · 2023-08-10
> > **Reviewer response**
> >
> > Thanks for your response. I appreciate the changes for the bolding in Table 1 and the clarification for the distinction between noisy and self-correctable datapoints, as well as the additional results on synthetic flips for ImageNet-1k.
> >
> > I'm happy to increase my score to a 5 but am not wholly convinced by the empirical results.

---

> > > ### Author Response · Authors · 2023-08-11
> > > **Thank you for your positive feedback**
> > >
> > > Thank you very much for your positive feedback! We are more than happy to answer any additional questions during the discussion period.  Moreover, we will keep polishing the evaluation section so that our contribution can be more clearly delivered.

---

### Official Review · Reviewer_jBWT · 2023-07-01

**Soundness:** 3 good
**Presentation:** 3 good
**Contribution:** 3 good
**Rating:** 5
**Confidence:** 5

**Summary:**

The paper proposes a novel data pruning algorithm, Prune4ReL, that maximizes the neighborhood confidence of the entire training examples, which is proportional to the likelihood of correct re-labeling. The paper demonstrates the effectiveness of Prune4ReL on four noisy datasets, where it outperforms baselines by a large margin.

**Strengths:**

The writing of this article is very clear and easy to follow up. Moreover, the methodology in this paper is also reasonable with necessary theoretical analysis. I enjoy this work very much! My only concern about this work is the experimental part.



**Weaknesses:**

1. Recent works like [1,2,3] are missing in those selected baselines.

2. Lack of results on ImageNet-1K, which is the most convincing part for us.

[1] Moderate: Moderate coreset: A universal method of data selection for real-world data-efficient deep learning. ICLR-2023

[2] SSP: “Beyond neural scaling laws: beating power law scaling via data pruning. NeurIPS-2022

[3] CCS: Coverage-centric Coreset Selection for High Pruning Rates. ICLR-2023

**Questions:**

In addition, the current dataset-pruning experiment mainly focuses on image classification, but this kind of task is too simple to reflect the real needs. For example, for CIFAR or even ImageNet, we don't need dataset-pruning at all, and the cost of training on these datasets is very affordable for almost all scientific teams. I think the real application scenario of dataset-pruning should be the most popular tasks such as large language model (LLM) training and multi-modal training. So, what do you think is the main difficulty of using dataset-pruning for these tasks? How is these tasks different from image classification?

At present, the performance of dataset-pruning algorithm in image classification tasks is generally not very good, for example, the performance degrades very seriously under high pruning-ratio Settings. Moreover, all existing pruning algorithms cannot outperform the ramdom selection baseline with a significant margin. However, in the field of LLM, as a contrast, some manually filtered datasets [1,2] can achieve a data set size reduction of tens of times, and performance remains unchanged or even significantly improved. So, can we say that manual filtering is still far better than algorithmic automated filtering? Or, what in the world is causing such a big gap.

[1]. Textbooks Are All You Need. Arxiv. [2306.11644]
[2]. LIMA: Less Is More for Alignment. Arxiv. [2305.11206]

**Limitations:**

Please refer to Weaknesses.

---

> ### Author Rebuttal · Authors · 2023-08-10
>
> `The writing of this article is very clear and easy to follow up. Moreover, the methodology in this paper is also reasonable with necessary theoretical analysis. I enjoy this work very much!`
> > We are very glad to hear that you enjoy reading our paper.
>
> `My only concern about this work is the experimental part.`
> > We did our best to improve the evaluation during the rebuttal period, and we hope that our efforts will address your sole concern.
>
>
> `Q1. Recent works like [a,b,c] are missing in those selected baselines.`
>
> Thank you very much for recommending important references. We have additionally conducted the experiments with Moderate[a] and SSP[b], and the results with SOP+ are reported as below. **Prune4Rel is also shown to outperform these recent baselines**, because they are not designed for the **noise-robust** learning scenario. We will add these results (also for other datasets) in the final version.
>
> | Methods        |         | CIFAR-10N | Random   |          |         | CIFAR-10N | Worst    |          |
> | -------------- | -------- | -------- | -------- | -------- | -------- | -------- | -------- | -------- |
> |                | 0.2      | 0.4      | 0.6      | 0.8      | 0.2      | 0.4      | 0.6      | 0.8      |
> | Uniform  | 87.5     | 91.5     | 93.4     | 94.8     | 81.9     | 87.5     | 90.8     | 91.8     |
> | SmallL   | 77.6     | 86.2     | 90.7     | 94.3     | 78.8     | 84.1     | 89.3     | 92.3     |
> | Margin   | 52.1     | 79.6     | 92.6     | 95.1     | 45.7     | 61.8     | 84.6     | **92.5** |
> | kCenter  | 86.3     | 92.2     | 94.1     | **95.3** | 81.9     | 88.0     | 91.3     | 92.3     |
> | Forget   | 82.4     | 93.0     | 94.2     | 95.0     | 71.1     | 87.7     | 90.6     | 92.2     |
> | GraNd    | 24.2     | 51.6     | 85.9     | 94.9     | 15.4     | 25.7     | 51.0     | 86.8     |
> | **Moderate** | 88.3     | 92.8     | 94.1     | 94.6     | 75.0     | 81.9     | 87.7     | 91.8     |
> | **SSP** | 80.5     | 91.7     | 93.8     | 95.0     | 70.8     | 86.6     | 89.2     | 92.3     |
> | **Pr4Rel**     | 88.2     | 93.0     | **94.4** | 95.1     | 83.4     | **89.3** | **91.5** | **92.5** |
> | **Pr4Rel$_B$** | **88.6** | **93.1** | 94.2     | **95.3** | **84.2** | 89.1     | 91.3     | **92.5** |
>
> We were unable to add CCS[c] within a week, but the results for CCS will be included in the final version.
>
> [a] Xiaobo Xia, Jiale Liu, Jun Yu, Xu Shen, Bo Han, Tongliang Liu: Moderate Coreset: A Universal Method of Data Selection for Real-world Data-efficient Deep Learning. ICLR 2023
>
> [b] Ben Sorscher, Robert Geirhos, Shashank Shekhar, Surya Ganguli, Ari Morcos: Beyond neural scaling laws: beating power law scaling via data pruning. NeurIPS 2022
>
> [c] Haizhong Zheng, Rui Liu, Fan Lai, Atul Prakash: Coverage-centric Coreset Selection for High Pruning Rates. ICLR 2023
>
>
> `Q2. Lack of results on ImageNet-1K, which is the most convincing part for us.`
>
> Thank you very much for helping us improve our paper. All datasets used for our experiments, CIFAR-10N, CIFAR-100N, WebVision, and Clothing-1M, have **real** label noises. However, for ImageNet-1K, we could not find its variant that contains real label noises. Thus, we have injected synthetic noises into ImageNet-1K. In detail, we flipped the original label (e.g., $classId = 0$) to an incorrect label (e.g., $classId = 1$) randomly by a given noise rate. The results for ImageNet-1K are provided as below and will be included in the final version. Here, the noise rate is set to be 20%, and SOP+ is trained for 50 epochs from scratch with a batch size of 64. Prune4Rel is shown to perform the best also in this dataset, thereby demonstrating its versatility. (More results for high selection ratios will come during the discussion phase.)
>
> | Methods         | ImageNet-1K | 20% Noise |     |
> | --------------- | ------- | -------- | -------- |
> | Selection Ratio | 0.01    | 0.05     | 0.1      |
> | Uniform         | 2.6     | 27.8     | 42.5     |
> | SmallL          | 5.8     | 22.8     | 31.4     |
> | Forget          | 0.8     | 4.1      | 8.3      |
> | **Pr4Rel$_B$**  | **6.0** | **30.2** | **44.3** |
>
>
> `Q3. I think the real application scenario of dataset-pruning should be the most popular tasks such as large language model (LLM) training and multi-modal training. So, what do you think is the main difficulty of using dataset-pruning for these tasks? How is these tasks different from image classification?`
>
> This is an excellent question. We agree with you that data pruning for multi-modal training and large language model (LLM) training will be more interesting than for image classification. Regarding LLM training, we believe that there are several challenges, compared to image classification. First, various levels---token, sentence, and document---can be considered as a pruning granularity for a large-scale corpus. Second, efficient selection criteria should be developed for LLMs, because most of the metrics typically used for image classification, e.g., gradients, are very expensive to calculate. Obviously, there will be additional challenges to investigate. After the review period concludes, we, the authors and the reviewer, can even attempt to collaborate on this extremely intriguing topic.
>
>
> `Q4. Can we say that manual filtering is still far better than algorithmic automated filtering? Or, what in the world is causing such a big gap?`
>
> Again, thank you for posing a very interesting research question. In our opinion, the quality, diversity, and redundancy of a given raw dataset are more important in determining the efficacy of data pruning than whether it is performed automatically or manually. (The capacity of a pre-trained model is also relevant given that [d] involves data pruning for fine-tuning.) Clearly, additional research is required to provide an exact answer to this question.
>
> [d] Chunting Zhou et al.: LIMA: Less Is More for Alignment. CoRR abs/2305.11206 (2023)

---

> > ### Comment · Reviewer_jBWT · 2023-08-14
> > **Thanks!**
> >
> > Congrats!
> >
> > I am very satisfied with your reply! This is a good article and very enlightening! I have a habit of giving conservative marks in the first stage review! I hope it will not affect you badly.

---

> > > ### Author Response · Authors · 2023-08-17
> > > **Thank you and more results on ImageNet-1K**
> > >
> > > We are happy to hear that you are very satisfied with our response. Also, even though the experiment for ImageNet-1K is not yet complete, we would like to update the progress using the interim results at this time. Prune4Rel is shown to maintain its dominance also in this dataset.
> > >
> > > | Methods         | ImageNet-1K | 20% Noise |     |  |  |
> > > | --------------- | ------- | -------- | -------- | -------- | -------- |
> > > | Selection Ratio | 0.01    | 0.05     | 0.1      | 0.2      | 0.4      |
> > > | Uniform         | 2.6     | 27.8     | 42.5     | 52.7     | 59.2     |
> > > | SmallL          | 5.8     | 22.8     | 31.4     | 42.7     | 54.4     |
> > > | Forget          | 0.8     | 4.1      | 8.3      | 50.6     | 57.2     |
> > > | **Pr4Rel$_B$**  | **6.0** | **30.2** | **44.3** | **53.5** | **60.0** |
> > >
> > >
> > > Please let us know if you have any additional questions. Again, your encouragement is greatly appreciated.

---

### Official Review · Reviewer_N3nf · 2023-07-06

**Soundness:** 4 excellent
**Presentation:** 4 excellent
**Contribution:** 3 good
**Rating:** 7
**Confidence:** 4

**Summary:**

The paper proposes Prune4ReL, which prunes a noisy training dataset such that the performance of a Re-labeling trained downstream model is maximized.
Unlike previous work, the paper targets pruning a *noisy* dataset and explicitly considers the learning algorithm of the downstream model. The proposed method is specifically tailored to Re-labeling, automatically cleaning the noisy training dataset.
The utility function in Prune4ReL is inspired by the theory proposed in [1] and gracefully incorporated into the context of data pruning.
The experimental results show that most prior work suffers under noisy datasets, while Prune4ReL remains robust to the noise in the training dataset, resulting in substantial improvements over baselines.

[1] Theoretical Analysis of Self-Training with Deep Networks on Unlabeled Data, ICLR2021

**Strengths:**

The strength of the paper include
- Clear presentation and easy-to-follow writing
- The proposed method is theoretically-inspired and, maybe more importantly, easy to implement
- The evaluation, together with the analysis, is convincing.
- Include the analysis on why and how the baseline fail

The paper is well-organized, and the method and evaluation are solid.

**Weaknesses:**

The paper conducts a complete study on the proposed Prune4ReL, and the following weakness is relatively minor.

- Prune4ReL outperforms the baselines by a large margin, but the gap between PruneReL and uniform sampling is small.
- Some notations are not clear.
  - Def. 3.1: the $x$ is not in the pixel space. Instead, it's in the embedding space. The author should state the dimension of $x$ at the beginning
  - in Thm 3.4, it should be $\mathcal{S} \subseteq \tilde{D}$

**Questions:**

Experiment
- Uniform sampling is usually the second-best baseline. It performs especially well in CIFAR-10N Random/Worst and Clothing-1M. Can the authors elaborate more on this?
  - Do the authors believe the gap between uniform sampling and Prune4ReL is small due to the greedy sampling? Or what are the other hypothesis?
- The derivation from the reduced neighbourhood confidence (Eq. 3) to the empirical reduced neighbour confidence (Eq.4) is confusing. What motivates using the cosine distance to perform a weighted sum? Is this motivated by importance sampling?
- WebVision and Clothing-1M are crawled from the web, but Prune4ReL seems to work better in Clothing-1M. Can the authors elaborate on this further?
  - It seems that kCenter and GraNd are missing in Figure 3b. The authors have to find a better way to visualize this plot (Figure 3b). Try to zoom in a little bit.
- Section 4.5 is like the motivation of the paper. Consider re-order them.

**Limitations:**

As mentioned in the #weakness, there are some minor limitations.
- Empirical results: The proposed method still requires some work on improving over uniform sampling baseline, and the authors did not justify the strong uniform baseline.

---

> ### Author Rebuttal · Authors · 2023-08-10
>
>
> We sincerely appreciate the reviewers' constructive comments and positive feedback on our manuscript.
>
> `Q1. Uniform sampling is usually the second-best baseline. It performs especially well in CIFAR-10N Random/Worst and Clothing-1M. Can the authors elaborate more on this?`
>
> To improve the performance of Re-labeling models, sample selection should achieve a balance between easy (clean) and hard (possibly noisy) examples. This is because the easy (clean) examples support the correct relabeling of neighbor hard examples, while the hard examples greatly boost the test accuracy if they are re-labeled correctly. Existing data pruning algorithms that prefer hard examples may perform poorly due to a lack of easy examples that support accurate relabeling, and clean sample selection algorithms that prefer easy examples may also perform poorly due to a lack of hard (informative) examples that aid in test accuracy after relabeling. On the other hand, uniform sampling shows somewhat robust performance by selecting easy and hard examples in a balanced way.
>
> For CIFAR-10N, as shown in the selection ratio of 0.2 in Table 1, uniform sampling was more effective at a small selection ratio because a proper amount of easy (clean) examples was collected to support accurate relabeling. However, looking at the selection ratio of 0.8, uniform sampling tended to show lower performance than other data pruning baselines since selecting hard examples became more important as there existed already proper amounts of easy examples. For Clothing-1M, we conjecture that uniform sampling performed well because a low selection ratio was used for testing the fine-tuning. Overall, we will add this discussion in the final version.
>
>
> `Q2. The derivation from the reduced neighborhood confidence (Eq. 3) to the empirical reduced neighborhood confidence (Eq.4) is confusing. What motivates using the cosine distance to perform a weighted sum? Is this motivated by importance sampling?`
>
> The reduced neighborhood confidence (Eq. 3) of each example is calculated as the sum of the confidences in its neighborhood. However, identifying the neighborhood of an example is very expensive and thus practically infeasible. Thus, instead of explicitly finding the neighborhood of each example, the *empirical* reduced neighborhood confidence (Eq. 4) approximates it using the cosine similarity **as the likelihood of belonging to the neighborhood**. We will elaborate on this rationale in the final version.
>
>
> `Q3. WebVision and Clothing-1M are crawled from the web, but Prune4ReL seems to work better in Clothing-1M. Can the authors elaborate on this further?`
>
> We interpret your question to mean that Prune4Rel achieves higher accuracy at lower selection ratios in Clothing-1M. Following the existing literature[a] for Clothing-1M, we used a ResNet model pretrained on ImageNet, as specified in Section 4.1. Thus, **fine-tuning was tested for Clothing-1M**, and this setting explains why high accuracy was achieved using a small portion of the Clothing-1M training set. Please inform us if we misunderstood your question.
>
> [a] Sheng Liu, Zhihui Zhu, Qing Qu, Chong You: Robust Training under Label Noise by Over-parameterization. ICML 2022: 14153-14172
>
>
> `Q4. kCenter and GraNd are missing in Figure 3b. The authors have to find a better way to visualize this plot (Figure 3b). Try to zoom in a little bit.`
>
> Thanks for pointing out this issue. The enhanced plot is included in the pdf file for the global response. Since *kCenter* requires huge computation and memory costs, it is not feasible to run in our environment (refer to Figure 3c). Also, since *Forgetting* requires prediction history during the warm-up training period to calculate the forgetting score of each training example, we could not run it for Clothing-1M where sample selection is performed from the pre-trained model.
>
>
> `Q5. Section 4.5 is like the motivation of the paper. Consider re-ordering them.`
>
> Thank you very much for helping us improve our paper. As you propely recognize, utilizing Re-labeling models for data pruning under label noise is much more effective than utilizing standard models. Thus, we will reorder Section 4.5 to Section 4.3 in order to confirm the motivation early on. In addition, we will add a summary of Section 4.5 at the end of Introduction: "When combined with the pruning methods, the performance of the Re-labeling models significantly surpasses that of the standard models in CIFAR-10N and CIFAR-100N by up to 21.6%."
>
>
> `Q6. Some notations are not clear.`
> > In Definition 3.1, the dimensionality of $x$ is set to be 512 for CIFAR-10N and CIFAR-100N and 2048 for WebVision and Clothing-1M. We will include this detail in the final version.
>
> > In Theorem 3.4, we will correct the notation to $\mathcal{S} \subseteq \tilde{\mathcal{D}}$. Thank you for pointing out the typo.

---

> > ### Comment · Reviewer_N3nf · 2023-08-13
> > **Reviewer response**
> >
> > Great work! The authors addressed all my concerns and please include the discussion of the uniform sampling baseline in the final version. I will keep my score.

---

> > > ### Author Response · Authors · 2023-08-14
> > > **Thank you**
> > >
> > > We are glad to hear that you are satisfied with our response and will surely incorporate this discussion into the final version. Thank you again for your support and insightful feedback.

---

### Author Rebuttal · Authors · 2023-08-10

We deeply appreciate the reviewers' positive feedback and valuable comments. Most reviewers agreed that (1) **the problem setting and methodology are reasonable and novel**, (2) **the theoretical analysis of the methodology is sound**, and (3) **the evaluation was performed extensively**.  Because the reviewer's comments are mostly about the experiments, **we have significantly improved (and are improving) the evaluation section** during the rebuttal period by adding two baselines, one dataset, and one metric.  (See the attached PDF file for an enhanced plot.)  Therefore, we believe that the superiority of Prune4Rel over other baselines has demonstrated much more clearly and hope that the remaining concerns are addressed by the rebuttal.  We are happy to answer additional questions during the discussion period.

---

### Decision · Program_Chairs · 2023-09-21

**Decision:**

Accept (poster)

**Comment:**

In the reviewers' words: this paper has a "interesting and well motivated" goal which is achieved by a "sound", "reasonable", "theoretically-inspired", and "easy to implement" algorithm, with "theoretical guarantees" and "convincing evaluation". "The writing of this article is very clear" with "clear presentation". Given the unanimous recommendation for acceptance from the four reviewers, I recommend acceptance.